# Learning Policy Committees for Effective Personalization in MDPs with Diverse Tasks

**Luise Ge** [1]  **Michael Lanier** [1]  **Anindya Sarkar** [1]  **Bengisu Guresti** [1]  **Chongjie Zhang** [1]  **Yevgeniy Vorobeychik** [1]

## Abstract

Many dynamic decision problems, such as robotic control, involve a series of tasks, many of which are unknown at training time. Typical approaches for these problems, such as multi-task and meta-reinforcement learning, do not generalize well when the tasks are diverse. On the other hand, approaches that aim to tackle task diversity, such as using task embedding as policy context and task clustering, typically lack performance guarantees and require a large number of training tasks. To address these challenges, we propose a novel approach for learning a *policy committee* that includes at least one near-optimal policy with high probability for tasks encountered during execution. While we show that this problem is in general inapproximable, we present two practical algorithmic solutions. The first yields provable approximation and task sample complexity guarantees when tasks are low-dimensional (the best we can do due to inapproximability), whereas the second is a general and practical gradient-based approach. In addition, we provide a provable sample complexity bound for few-shot learning. Our experiments on MuJoCo and Meta-World show that the proposed approach outperforms state-of-the-art multi-task, meta-, and task clustering baselines in training, generalization, and few-shot learning, often by a large margin. Our code is available at https://github.com/CERL-WUSTL/PACMAN/.

## 1. Introduction

Reinforcement learning (RL) has achieved remarkable success in a variety of domains, from robotic control (Lilli-

[1]Department of Computer Science & Engineering, Washington University in St. Louis. Correspondence to: Luise Ge <g.luise@wustl.edu>.

*Proceedings of the 42$^{nd}$ International Conference on Machine Learning*, Vancouver, Canada. PMLR 267, 2025. Copyright 2025 by the author(s).

crap, 2015) to game playing (Xu et al., 2018). However, many real-world applications involve highly diverse sets of tasks, making it impractical to rely on a single, fixed policy. In these settings, both the reward structures and the transition dynamics can vary significantly across tasks. Existing approaches, such as multi-task RL (MTRL) and meta-reinforcement learning (meta-RL), struggle to generalize effectively for diverse and previously unseen tasks.

Multi-task RL methods typically train a single policy or a shared representation across tasks (Vithayathil Varghese & Mahmoud, 2020). However, they often face negative transfer, where optimizing for one task degrades performance on others (Zhang et al., 2022). On the other hand, meta-RL approaches, such as Model-Agnostic Meta-Learning (MAML) (Finn et al., 2017b) and PEARL (Rakelly et al., 2019), aim to enable fast adaptation to new tasks but rely heavily on fine-tuning at test time, which can be computationally expensive and ineffective in environments with high variability in both rewards and transitions, like Meta-World benchmark tasks (Yu et al., 2020b).

Several lines of work attempt to address the problem of diverse tasks and negative transfer. The first is to learn policies through MTRL or meta-RL that explicitly take task representation as input (Lan et al., 2024; Grigsby et al., 2024; Sodhani et al., 2021; Zintgraf et al., 2020). However, such approaches rely on shared parameters, which limit the model's flexibility. When tasks are highly diverse, using a single neural network— even with a multi-head architecture— may not generalize effectively. Moreover, the greater the task variation, the more data is typically required to train policies that can effectively condition on task embeddings, and the model will often generalize poorly when the number of training tasks is small.

Several alternative methods have thus emerged that propose clustering tasks and training distinct policies for each cluster (Ackermann et al., 2021; Ivanov & Ben-Porat, 2024). The associated algorithmic approaches commonly leverage EM-style methods that interleave RL and task clustering. In practice, however, they also require a large number of training tasks to be effective and consequently perform poorly when the number of training tasks is small. In addition, if the number of clusters is too small, some clusters may still

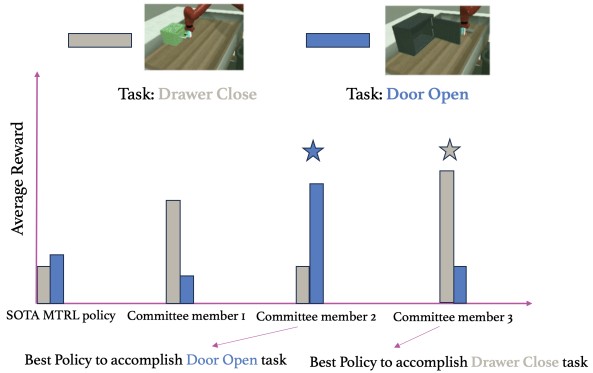

*Figure 1.* Performance on a single task across committee members compared to a MTRL policy.

contain too diverse a set of tasks, with concomitant negative transfer remaining a significant issue.

We propose PACMAN, a novel framework and algorithmic approach for learning **policy committees** that enables (a) sample efficient generalization (Theorem 3.7), (b) few-shot adaptation guarantees that are *independent of the dimension of the state or action space* (Theorem 3.8), and (c) significant improvements in performance compared to 11 state-of-the-art baselines on both MTRL and few-shot adaptation metrics (Section 4). The key idea behind PACMAN is to leverage a parametric task representation which allows clustering tasks in the parameter space first, followed by reinforcement learning for each cluster. The benefit of our approach is illustrated in Figure 1: even though the single-policy baseline uses a mixture-of-experts approach (Hendawy et al., 2024), its performance is poor since tasks require fundamentally distinct skills. In contrast, learning a policy committee allows learning custom policies for distinct classes of tasks.

A crucial insight in our approach is to define the objective of clustering as obtaining *high coverage* of the (unknown) task distribution, as opposed to *insisting on a full coverage*—that is, achieving near-optimal performance on a randomly drawn task with high probability. This enables us to devise efficient algorithms with provable performance guarantees and polynomial task sample complexity, as well as obtain few-shot adaptation guarantees that depends only on the number of clusters, but not on the size of state and action space. Moreover, while many MTRL and meta-RL problems of interest are non-parametric, we show that we can leverage high-level natural language task descriptions to obtain highly effective embeddings of tasks, akin to Sodhani et al. (2021), and Hendawy et al. (2024), but without requiring subtask decomposition. Consequently, our approach can be effectively applied to a broad class of non-parametric MTRL and meta-RL domains as well.

In summary, we make the following contributions:

- **Theoretically Grounded Framework for Learning Policy Committees:** We present the first approach for learning policy committees in MTRL and meta-RL domains that offers provable performance guarantees, in addition to practical algorithms. In particular, we provide sample efficient generalization (Theorem 3.7) and few-shot adaptation guarantees that are independent of the size of the state or action space (Theorem 3.8).

- **Simple and Effective Policy Committees**: PACMAN is design to handle high task diversity while using a small number of policies to achieve high efficacy. Additionally, our approach leverages LLM-based task embeddings for non-parametric tasks, which provides a highly general and scalable solution for a broad array of environments.

- **Empirical Validation:** We demonstrate the efficacy of PACMAN through extensive experiments on challenging multi-task benchmarks, including MuJoCo and Meta-World. Our policy committee framework consistently outperforms state-of-the-art multi-task RL, meta-RL, and task clustering baselines in both zero-shot (MTRL) and few-shot (meta-RL) settings, achieving better generalization and faster adaptation across diverse tasks.

**Related Work:** Our work is closely related to three key areas within the broader reinforcement learning literature: multi-task RL, meta-RL, and personalized RL.

*Multi-Task RL (MTRL):* The broad goal of MTRL approaches is to leverage inter-task relationships to learn policies that are effective across multiple tasks (Yang et al., 2020; Sodhani et al., 2021; Sun et al., 2022). An important challenge in MTRL is task interference. One class of approaches aims to mitigate this issue through gradient alignment techniques (Hessel et al., 2019; Yu et al., 2020a). An alternative series of approaches addresses this issue from a representation learning perspective, enabling the learning of policies that explicitly condition on task embeddings (Hendawy et al., 2024; Lan et al., 2024; Sodhani et al., 2021).

However, most MTRL methods still face challenges when tasks are diverse, particularly when it comes to generalizing to *previously unseen tasks*.

*Meta-RL:* The goal of meta-RL is an ability to quickly adapt to unseen tasks—what we refer to as *few-shot learning*. Meta-RL methods can be categorized broadly into two categories: (i) gradient-based and (ii) context-based (where context may include task-specific features). Gradient-based approaches focus on learning a shared initialization of a model across tasks that is explicitly trained to facilitate few-shot learning (Finn et al., 2017b; Stadie et al., 2018; Mendonca et al., 2019; Zintgraf et al., 2019). However, they perform poorly in zero-shot generalization, and tend to require a large number of adaptation steps. Context-based methods learn a context representation and use it as a policy

input (Bing et al., 2023; Gupta et al., 2018; Duan et al., 2017; Lee et al., 2020a;b; 2023; Rakelly et al., 2019). However, these approaches often exhibit mode-seeking behavior and struggle to generalize, particularly when the number of training tasks is small. While some recent approaches, such as Bing et al. (2023), attempt to improve performance by using natural language task embedding, they still require a large number of training tasks to succeed.

*Personalized RL:* Ackermann et al. (2021) and Ivanov & Ben-Porat (2024) proposed addressing task diversity by clustering RL tasks and training a distinct policy for each cluster. Both use EM-style approaches to jointly cluster the tasks and learn a set of cluster-specific policies. Our key contribution is to leverage an explicit parametric task representation, and reformulate the objective as a flexible *coverage* problem for an unknown distribution of tasks. This enables us to achieve task sample efficiency both in theory and practice. In particular, we empirically show that our PACMAN method significantly outperforms the state-of-the-art personalized RL approach.

*Reward-Free RL:* Another related line of research is reward-free RL where the agent's goal is to first explore the MDP without a pre-specified reward function and then plan near-optimal policies for a set of given reward functions (Agarwal et al., 2023; Cheng et al., 2022; Jin et al., 2020). However, results in this space make strong assumptions, such as assuming that the action space is finite or linear dynamics. Our results, on the other hand, make no assumptions on the dynamics and do not depend on the size or dimension of the state or action space.

## 2. Model

We consider the following general model of *multi-task MDPs (MT-MDP)*. Suppose we have a *dynamic environment* $\mathcal{E} = (\mathcal{S}, \mathcal{A}, h, \gamma, \rho)$ where $\mathcal{S}$ is a state space, $\mathcal{A}$ an action space, $h$ the decision horizon, $\gamma$ the discount factor, and $\rho$ the initial state distribution. Let a *task* $\tau = (\mathcal{T}, r)$ in which $\mathcal{T}$ is the transition model where $\mathcal{T}(s, a)$ is a probability distribution over next state $s'$ as a function of current state-action pair $(s, a)$ and $r(s, a)$ the reward function. A Markov decision process (MDP) is thus a composition of the dynamic environment and task, $(\mathcal{E}, \tau)$.

Let $\Gamma$ be a distribution over tasks $\tau$. We define a *MT-MDP* $\mathcal{M}$ as the tuple $(\mathcal{E}, \Gamma)$, as in typical meta-RL models (Beck et al., 2023; Wang et al., 2024). Additionally, we define a *finite-sample* variant of MT-MDP, *FS-MT-MDP*, as $\mathcal{M}_n = (\mathcal{E}, \tau_1, \ldots, \tau_n)$, where $\tau_i \sim \Gamma$. An FS-MT-MDP thus corresponds to multi-task RL (Zhang & Yang, 2021).

At the high level, our goal is to learn a *committee of policies* $\Pi$ such that for most tasks, there exists at least one policy

$\pi \in \Pi$ that is effective.[1] Next, we formalize this problem.

Let $V_\tau^\pi$ be the value of a policy $\pi$ for a given task $\tau$, i.e.,

$$V_\tau^\pi = \mathbb{E}\left[\sum_{t=0}^{h} \gamma^t r_\tau(s_t, a_t) | a_t = \pi(s_t)\right],$$

where the expectation is with respect to $\mathcal{T}_\tau$ and $\rho$. Let $\mathcal{P}$ be a (possibly restricted, e.g., parametric) space of policies that we take to be exogenously specified. We define $V_\tau^* = \max_{\pi \in \mathcal{P}} V_\tau^\pi$ as the optimal value for a task $\tau$, that is, the value of an optimal policy for $\tau$.

Define $V_\tau^\Pi = \max_{\pi \in \Pi} V_\tau^\pi$, that is, we let the value of a committee $\Pi$ to a task $\tau$ be the value of the *best* policy in the committee for this task. There are a number of reasons why this evaluation of a committee is reasonable. As an example, if a policy implements responses to prompts for conversational agents and $\Pi$ is small, we can present multiple responses if there is significant semantic disagreement among them, and let the user choose the most appropriate. In control settings, we can rely on domain experts who can use additional semantic information associated with each $\pi \in \Pi$ and the tasks, such as the descriptions of tasks $\pi$ was effective for at training time, and similar descriptions to test-time tasks, to choose a policy. Moreover, as we show in Section 3.4, this framework leads naturally to effective few-shot adaptation, which requires neither user nor expert input to determine the best policy.

One way to define the value of a policy committee $\Pi$ with respect to a given MT-MDP and FS-MT-MDP is, respectively, as $V_{\mathcal{M}}^\Pi = \mathbb{E}_{\tau \sim \Gamma}\left[V_\tau^\Pi\right]$ and $V_{\mathcal{M}_n}^\Pi = \frac{1}{n}\sum_{i=1}^{n} V_{\tau_i}^\Pi$. The key problem with these learning goals is that when the set of tasks is highly diverse, different tasks can confound learning efficacy for one another. For example, suppose that we have five tasks corresponding to target velocities of $10, 12, 20, 22, 100$, and the task succeeds if a policy implements its target velocity sufficiently closely (say, within 1). If we either train a single policy for all tasks, or divide them into two clusters, the outlier target velocity of 100 will confound training for the others. More generally, if any policy is trained on a set of tasks that require different skills (e.g., a cluster of tasks that includes outliers), the conflicting reward signals will cause negative transfer and poor performance.

We address this limitation by defining the goal of policy committee learning differently. First, we formalize what it means for a committee $\Pi$ to have a *good* policy for *most* of the tasks.

**Definition 2.1.** A policy committee $\Pi$ is an $(\epsilon, 1 - \delta)$-*cover* for a *task distribution* $\Gamma$ if $V_\tau^\Pi \geq V_\tau^* - \epsilon$ with probability at least $1 - \delta$ with respect to $\Gamma$. $\Pi$ is an $(\epsilon, 1 - \delta)$-*cover* for

---

[1]Note that this also admits policies that can further depend on task embeddings, with this dependence being different in different clusters.

*a set of tasks* $\{\tau_1, \ldots, \tau_n\}$ if $V_\tau^\Pi \geq V_\tau^* - \epsilon$ for at least a fraction $1 - \delta$ of tasks.

Clearly, an $(\epsilon, 1 - \delta)$ cover need not exist for an arbitrary committee $\Pi$ (if the committee is too small to cover enough tasks sufficiently well). There are, however, three knobs that we can adjust: $K$, $\epsilon$, and $\delta$. Next, we fix $\epsilon$ as exogenous, treating it effectively as a domain-specific hyperparameter, and suppose that $K$ is a pre-specified bound on the maximum size of the committee.

**Problem 1.** *Fix the maximum committee size $K$ and $\epsilon$. Our goal is to find $\Pi$ which is a $(\epsilon, 1 - \delta)$-cover for the smallest $\delta \in [0, 1]$.*

As a corollary to Theorem 2.6 in Skalse & Abate (2023), we now present sufficient conditions under which a policy committee can yield higher expected rewards than a single policy. The precise definition of these conditions is provided in Appendix A.

**Corollary 2.2** (Theorem 2.6, (Skalse & Abate, 2023))**.** *Suppose $K > 1$. Unless two tasks $\tau_1, \tau_2$ have only their reward functions $r_1$ and $r_2$ differ by potential shaping, positive linear scaling, and $S_1$-redistribution, we have $V_{\tau_1}^\Pi \geq V_{\tau_1}^\pi$, $V_{\tau_2}^\Pi \geq V_{\tau_2}^\pi$, and $V_{\tau_1}^\Pi + V_{\tau_2}^\Pi > V_{\tau_1}^\pi + V_{\tau_2}^\pi$ for any single policy $\pi$.*

Next, we present algorithmic approaches for this problem. Subsequently, Section 3.4, as well as our experimental results, vindicate our choice of this objective.

# 3. Algorithms for Learning a Policy Committee

In this section, we present algorithmic approaches for train committee members to solve Problems 1. We consider the special case of the problem in which the tasks have a *structure representation*. Specifically, we assume that each task can be represented using a parametric model $\psi_\theta(s, a)$, where the parameters $\theta \in \mathbb{R}^d$ comprise both of the parameters of the transition distribution $\mathcal{T}$ and reward function $r$. Often, parametric task representation is given or direct; in cases when tasks are non-parametric, such as the Meta-World (Yu et al., 2020b), we can often use approaches for task embedding, such as LLM-based task representations (see Section 3.6). Consequently, we identify tasks $\tau$ with their representation parameters $\theta$ throughout, and overload $\Gamma$ to mean the distribution over task parameters, i.e., $\theta \sim \Gamma$.

## 3.1. A High-Level Algorithmic Framework

Even conventional RL presents a practical challenge in complex problems, as learning is typically time consuming and requires extensive hyperparameter tuning. Consequently, a crucial consideration in algorithm design is to minimize the number of RL runs we need to obtain a policy committee.

To this end, we propose the following high-level algorithmic framework in which we only need $K$ independent (and, thus, entirely *parallelizable*) RL runs. This framework involves three steps:

1. SAMPLE $n$ tasks i.i.d. from $\Gamma$, obtaining $T = \{\theta_1, \ldots, \theta_n\}$ (parameters of associated tasks $\{\tau_1, \ldots, \tau_n\}$). In MTRL settings, $T$ is given.

2. CLUSTER the task set $T$ into $K$ subsets, each with an associated representative $\theta_k$, and

3. TRAIN committee member $c_k \in \Pi$ for each cluster $k$ represented by $\theta_k$.

As we shall see presently (and demonstrate experimentally in both Subsection 4.3 and Appendix H.2), conventional clustering approaches are not ideally suited for our problem. We thus propose several alternative approaches which yield theoretical guarantees on the quality of $\Pi$, in the special case that each committee member represents one policy.

Empirically, we show that the proposed framework outperforms state of the art even when tasks also have distinct transition distributions.

## 3.2. Clustering

The key aspect of our algorithmic design is clustering. We are motivated by a connection between the clustering step (step (2) of the framework above) and efficacy of optimal policies learned for each cluster (step (3) of the framework), as a variant of simulation lemma (Lobel & Parr, 2024), whose formal implications are discussed in the next section. By leveraging a strong representation space, we can avoid the overhead of evaluating and clustering tasks on-policy and achieve high efficacy with minimal computational cost. Thus, we focus on the following problem instead:

**Definition 3.1.** A set of representatives $C = \{\theta_1, \ldots, \theta_K\}$ is an $(\epsilon, 1 - \delta)$-*parameter-cover* for a *task distribution* $\Gamma$ if $\min_{\theta' \in C} \|\theta - \theta'\|_\infty \leq \epsilon$ with probability at least $1 - \delta$ with respect to $\theta \sim \Gamma$.

**Problem 2.** *Fix $K$ and $\epsilon$. Our goal is to find $C$ with $|C| \leq K$ which is a $(\epsilon, 1 - \delta)$-parameter-cover for the smallest $\delta \in [0, 1]$.*

Notably, while conventional clustering techniques, such as k-means, can be viewed as proxies for these objectives, there are clear differences insofar as the typical goal is to minimize sum of shortest distances of *all* vectors from cluster representatives, whereas our goal, essentially, is to "cover" as many vectors as we can. In Appendix H.2, we provide a histogram of individual task returns to illustrate the different impact the two clustering methods make.

We show next that our problem is strongly inapproximable, *even if we restrict attention to $K = 1$.*

**Definition 3.2** (MAX-1-COVER). Let $T = \{\theta_1, \ldots, \theta_n\} \subseteq \mathbb{R}^d$. Find $\theta \in \mathbb{R}^d$ which maximizes the size of $S \subseteq T$ with $\max_{\theta' \in S} \|\theta - \theta'\|_\infty \leq \epsilon$.

**Theorem 3.3.** *For any constant $t > 0$, MAX-1-COVER does not admit an $n^{1-t}$-approximation unless P = NP.*

We prove this in Appendix B via an approximation-preserving reduction from the Maximum Clique problem (Engebretsen & Holmerin, 2000). Despite this strong negative result, we next design two effective algorithmic approaches. The first method runs in polynomial time with a constant $d$, and provides a constant-factor approximation. The second is a general gradient-based approach.

**Greedy Elimination Algorithm (GEA)** Before we discuss our main algorithmic approaches, we begin with GEA, which provides a useful building block, but not theoretical guarantees. Consider a set $T$ of task parameter vectors, fix $K$, and suppose we wish to identify an $(\epsilon, 1 - \delta)$-parameter-cover with the smallest $\delta$ (Problem 2), but restrict attention to $\theta \in T$ in constructing such a cover. This problem is an instance of a MAX-K-COVER problem (where subsets correspond to sets covered by each $\theta \in T$), and can be approximated using a greedy algorithm which iteratively adds one $\theta \in T$ to $C$ that maximizes the most uncovered vectors in $T$. Its fixed-$\delta$ variant, on the other hand, is a set cover problem if $\delta = 0$, and a similar greedy algorithm approximates the minimum-$K$ cover $C$ for any $\delta$. However, neither of these algorithms achieves a reasonable approximation guarantee (as we can anticipate from Theorem 3.3), although our experiments show that greedy elimination is nevertheless an effective heuristic. But, as we show next, we can do better if we allow cluster covers $\theta$ to be unrestricted.

**Greedy Intersection Algorithm (GIA)** We now present our algorithm which yields provable approximation guarantees when task dimension is low. The key intuition behind GIA is that for any $\theta$, a $\epsilon$-hybercube centered at $\theta$ characterizes all possible $\theta'$ that can cover $\theta$ in the sense of Definition 3.1. Thus, if any pair of $\epsilon$-hypercubes centered at $\theta$ and $\theta'$ intersects, any point at the intersection covers both. To illustrate, consider the following simple example:

$$[x_3, x_4] \qquad\qquad\qquad x_4 - \epsilon \;\dashv\!\!\times\!\!\longrightarrow\; x_4 + \epsilon$$
$$[x_1, x_2, x_3] \qquad\quad x_3 - \epsilon \;\longmapsto\!\!\times\!\!\vdash\; x_3 + \epsilon$$
$$[x_1, x_2] \qquad\quad x_2 - \epsilon \;\longrightarrow\!\!\times\!\!\vdash\; x_2 + \epsilon$$
$$[x_1] \qquad x_1 - \epsilon \;\longrightarrow\!\!\times\!\!\vdash\; x_1 + \epsilon$$

Each cross represents a parameter we aim to cover, while each line segment indicates the possible locations of the $\epsilon$-close representative for that parameter. By selecting a point within the overlapping region of these intervals, we can effectively cover their parameters simultaneously.

The proposed GIA algorithm generalizes this intuition as

follows. The first stage of the algorithm is to create an intersection tree for each dimension independently. For $s$-th dimension, we sort the datapoints' $s$-th coordinates in ascending order. We refer to the sorted coordinates as $x_1 < x_2 \cdots < x_n$, and create a list for each point $x_i$ to remember how many other points can be covered together with it with initialization being $[x_i]$ itself.

Starting from the second smallest datapoint $x_2$, we check if $x_2 - \epsilon \leq x_1 + \epsilon$, i.e. if $x_2 \leq x_1 + 2\epsilon$. Since $x_2 - \epsilon > x_1 - \epsilon$ due to our sorting, any point inside $[x_2 - \epsilon, x_1 + \epsilon]$ can cover both $x_1, x_2$. Therefore if this interval is valid, we add $x_1$ to the list $[x_2]$ to indicate the existence of a simultaneous coverage for $x_1, x_2$. In general, for $x_i$, we check if $x_i \leq x_j + 2\epsilon$ with a descending $j = i - 1$ to 1 or until the condition no longer holds. If the inequality is satisfied, we add $x_j$ to $x_i$'s list. Then since we have ordered the set, for every index $j'$ less than $j$, $x_i > x_j + 2\epsilon > x_{j'} + 2\epsilon$. The coverage for all the $x$ in $x_i$'s list would be the interval $[x_i - \epsilon, x_j + \epsilon]$, where $j$ is the smallest index in $x_i$'s list. There are $1 + 2 + \cdots + n - 1 = \mathcal{O}(n^2)$ comparisons in total. We form a set of these lists, and call it $\mathcal{A}_s$ for the $s$-th dimension. The figure above illustrates how the algorithm works to find out $\mathcal{A}_1 = \{[x_1], [x_1, x_2], [x_1, x_2, x_3], [x_4, x_5]\}$.

The second stage is to find a hypercube covering the most points, consisting of an axis from each dimension. By the geometry of the Euclidean space, two points $\theta_1, \theta_2$ are within $\epsilon$ in $\ell_\infty$-distance iff they appear inside one's list together for each dimension. Therefore, in order to find the maximum coverage with one hypercube such that its center is within $\ell_\infty$-distance to the most points, we wonder which combination of lists, $l_1 \ldots l_d$ each from the sets $\mathcal{A}_1 \ldots \mathcal{A}_d$ produces an intersection of the maximum cardinality. In our example, we can conclude that $[x_1, x_2, x_3], [x_4, x_5]$ need to be covered separately by two points between the blue or red vertical lines.

The full algorithm is provided in Appendix C. Next, we show that GIA yields provable coverage guarantees. We defer the proof to Appendix D. For these results we use GIA($K$) to refer to the solution (set $C = \{\theta_1, \ldots, \theta_K\}$) returned by GIA.

**Theorem 3.4.** *Suppose $T$ contains $n \geq \frac{9 \log(5/\alpha)}{2\beta^2}$ i.i.d. samples from $\Gamma$. Let $1 - \delta^*(K, \epsilon)$ be the optimal coverage of a $\epsilon$-parameter-cover of $\Gamma$ under fixed $K, \epsilon$. Then with probability at least $1 - \alpha$, GIA($K$) is a $(\epsilon, (1 - \frac{1}{e})(1 - \delta^*(K, \epsilon) - K\beta))$-parameter-cover of $\Gamma$.*

The key limitation of GIA is that it is exponential in $d$, and thus requires the dimension to be constant. This is a reasonable assumption in some settings, such as low-dimensional control. However, in many other settings, both $n$ and $d$ can be large. Our next algorithm addresses this issue.

**Gradient-Based Coverage** Consider Problem 2. For a

finite set $T$, we can formalize this as the following optimization problem:

$$\max_{\{\theta_1,\ldots,\theta_K\}} \sum_{\theta \in T} \mathbf{1}\left(\min_{k \in [K]} \|\theta_k - \theta\|_\infty \leq \epsilon\right), \quad (1)$$

where $\mathbf{1}(\cdot)$ is 1 if the condition is true and 0 otherwise. However, objective (1) is non-convex and discontinuous. To address this, we propose the following differentiable proxy:

$$\min_{\substack{\{\theta_1,\ldots,\theta_K\}; \\ w \in \mathbb{R}^{nK}}} \sum_i \mathbf{ReLU}\left(\left\{\sum_{k=1}^K \sigma(w_{ik})\|\theta_k - \theta_i\|_\infty\right\} - \epsilon\right), \quad (2)$$

where $w$ is the assignment matrix mapping each $\theta_i$ to $\theta_k$ with unbounded weights and $\sigma(\cdot)$ is a softmax function to normalize the assignment. Next, we demonstrate that this is a principled proxy by showing that when full coverage of $T$ is possible, solutions of (1) and (2) coincide. The proof is in Appendix E.

**Theorem 3.5.** *Fix $K$ and suppose $\exists \theta \in \{\theta_1, \ldots, \theta_K\}$ such that $\|\theta - \theta_i\| \leq \epsilon$ for all $i$. Then the sets of optimal solutions to (1) and (2) are equivalent.*

Thus, we can use gradient-based methods with objective in (2) to approximate solutions to Problem 2. Because the objective is still non-convex, we can improve performance by initializing with the solution obtained using GEA or GIA when $d$ is low.

### 3.3. Theoretical Analysis for MTRL

We now put things together by showing that we have a sample efficient approach for learning a policy committee $\Pi$ which achieves a $(\epsilon, 1 - \delta)$-cover for $\Gamma$. For this result, we focus on the special case where each committee member is one fixed policy and assume that each task has a shared dynamics, and a parametric reward function $r_\theta(s, a)$ where $\theta$ identifies a task-specific reward. While this is a theoretical limitation, we note that our subsequent clustering and training algorithms do not in themselves require this assumption, and our experimental results demonstrate that the overall approach is effective generally.

Let $\pi_i^*$ denote the optimal policy for task $\tau_i$. We use $V_i^{\pi_j^*}$ to denote the value of task $\tau_i$ using a policy that is optimal for task $\tau_j$.

**Lemma 3.6.** *Suppose that $r_\theta(s, a)$ is $L$-Lipschitz in $L_\infty$ norm, that is, for all $\theta, \theta'$, $\sup_{s,a} |r_\theta(s, a) - r_{\theta'}(s, a)| \leq L\|\theta - \theta'\|_\infty$. Then, for any two tasks $\tau_i$ and $\tau_j$ with respective $\theta_i$ and $\theta_j$ that satisfy $\|\theta_i - \theta_j\|_\infty \leq \epsilon$, $V_i^{\pi_j^*} \geq V_i^{\pi_i^*} - 2L\frac{1-\gamma^{h+1}}{1-\gamma}\epsilon$ if $\gamma < 1$ and $V_i^{\pi_j^*} \geq V_i^{\pi_i^*} - 2Lh\epsilon$ if $\gamma = 1$.*

See Appendix F for proof. Lipschitz continuity is a mild assumption; for example, it is satisfied by ReLU neural networks.

Next, we combine Theorem 3.4 and Lemma 3.6 to conclude that we can compute a policy committee $\Pi$ which provides provable coverage guarantees for a task distribution $\Gamma$ with polynomial sample complexity.

**Theorem 3.7.** *Suppose $T$ contains $n \geq \frac{9\log(5/\alpha)}{2\beta^2}$ i.i.d. samples from $\Gamma$ and $r_\theta(s, a)$ is $L$-Lipschitz in $L_\infty$. Let $1 - \delta^*(K, \epsilon)$ be the optimal coverage of a $\epsilon$-parameter-cover of $\Gamma$ under fixed $K, \epsilon$, and suppose that for any task $\tau$ we can find a policy $\pi$ with $V^\pi \geq V_\tau^* - \eta$. Then we can compute a committee $\Pi$ which is a $(2L\frac{1-\gamma^{h+1}}{1-\gamma}\epsilon + \eta, (1 - \frac{1}{e})(1 - \delta^*(K, \epsilon) - K\beta))$ for $\Gamma$ when $\gamma < 1$ and $(2Lh\epsilon + \eta, (1 - \frac{1}{e})(1 - \delta^*(K, \epsilon) - K\beta))$ for $\Gamma$ when $\gamma = 1$ with probability at least $1 - \alpha$.*

### 3.4. Few-Shot Adaptation

A particularly useful consequence of learning a policy committee $\Pi$ that is a $(\epsilon, 1 - \delta)$-cover is that we can leverage it in meta-learning for few-shot adaptation. The algorithmic idea is straightforward: evaluate each of $K$ policies in $\Pi$ by computing a sample average sum of rewards over $N$ randomly initialized episodes, and choose the best policy $\pi \in \Pi$ in terms of empirical average reward.

In particular, suppose that $\gamma = 1$. We now show that this translates into a few-shot sample complexity on a previously unseen task $\tau$ that is linear in $K$ (the size of the committee). Details of the proof are in Appendix G.

**Theorem 3.8.** *Suppose $\Pi$ is a $(\epsilon, 1 - \delta)$-cover for $\Gamma$ and let $\tau \sim \Gamma$. Under some mild conditions, if we run $p \geq \frac{32h(H+1)^2 \log(4/\alpha)}{(\beta - 2H)^2}$ episodes for each policy $\pi \in \Pi$, where $H$ is a constant, the policy $\pi$ that maximizes the empirical return yields $V_\tau^\pi \geq V_\tau^* - \epsilon - \beta$ with probability at least $1 - \delta - \alpha$, where $V_\tau^*$ is the optimal reward for $\tau$.*

### 3.5. Training

The output of the CLUSTERING step above is a set of representative task parameters $C = \{\theta_1, \ldots, \theta_K\}$. The simplest way to use these to obtain a policy committee $\Pi$ is to train a policy $\pi_k$ optimized for each $\theta_k \in C$. However, this ignores the set of tasks that comprise each cluster $k$ associated with a representative $\theta_k$ (i.e., the set of tasks closest to $\theta_k$). As demonstrated empirically in the multi-task RL literature, using multiple tasks to learn a shared representation facilitates generalization (effectively enabling the model to learn features that are beneficial to all tasks in the cluster) (Sodhani et al., 2021; Sun et al., 2022; Yang et al., 2020).

To address this, we propose an alternative which trains a policy $\pi_k$ to maximize the sum of rewards of the tasks in

cluster $k$. Notably, our approach can use *any* RL algorithm to learn a policy associated with a cluster of tasks; in the experiments below, we use the most effective MTRL or meta-RL baseline for this purpose.

Furthermore, we emphasize that the additional overhead introduced by our method (due to clustering) is small, with the computational complexity in practice being predominantly determined by the RL problem. To illustrate, our Meta-World experiments show that training a single policy for 1 million steps necessitates approximately 40 hours using an A40 GPU. Conversely, the clustering process completes in roughly 1 second within a Google Colab notebook, and obtaining task embeddings takes approximately 2 minutes on an A40 GPU. Importantly, in cases where efficient RL learning is achievable (Brafman & Tennenholtz, 2002; Kearns & Singh, 2002) and the dimension d remains constant, our approach additionally boasts polynomial algorithmic complexity.

### 3.6. Dealing with Non-Parametric Tasks

Our approach assumes that tasks are parametric, so that we can reason (particularly in the clustering step) about parameter similarity. Many practical multi-task settings, however, are non-parametric, so that our algorithmic framework cannot be applied directly. In such cases, our approach can make use of any available method for extracting a parametric representation of an arbitrary task $\tau$. For example, it is often the case that tasks can be either described in natural language. We propose to leverage this property and use text embedding (e.g., from pretrained LLMs) as the parametric representation of otherwise non-parametric tasks, where this is feasible. Our hypothesis is that this embedding captures the most relevant semantic aspects of many tasks in practice, a hypothesis that our results below validate in the context of the Meta-World benchmark. This is analogous to what was done by Bing et al. (2023), with the main difference being that our task descriptions are with respect to higher-level goals, whereas Bing et al. (2023) describe tasks in terms of associated plans. We provide the full list of task descriptions for the Meta-World environment in Appendix I.

## 4. Experiments

We study the effectiveness of our approach—PACMAN—in two environments, *MuJoCo* (Todorov et al., 2012) and *Meta-World* (Yu et al., 2020b). In the former, the tasks are low-dimensional and parametric, and we only vary the reward functions, whereas the latter has non-parametric robotic manipulation tasks with varying reward and transition dynamics.

**MuJoCo** We selected two commonly used MuJoCo environments. The first is HalfCheetahVel where the agent has

to run at different velocities, and rewards are based on the distance to a target velocity. The second is HumanoidDir where the agent has to move along the preferred direction, and the reward is the distance to the target direction. In both, we generate diverse rewards by randomly generating target velocity and direction, respectively, and use 100 tasks for training and another 100 for testing (in both zero-shot and few-shot settings), with parameters generated from a Gaussian mixture model with 5 Gaussians. In few-shot cases, we draw a single task for fine-tuning, and average the result over 10 tasks. For clustering, we use $K = 3, \epsilon = .6$, and use the gradient-based approach initialized with the result of the *Greedy Intersection* algorithm. For few-shot learning, we fine-tune all methods for 100 epochs.

**Meta-World** We focus on the set of robotic manipulation tasks in MT50, of which we use *30 for training and 20 for testing*. This makes the learning problem significantly more challenging than typical in prior MTRL and meta-RL work, where training sets are much larger compared to test sets (5 tests and 40 trains in the traditional MT45 setting). We leverage an LLM to generate a parameterization (Section 3.6) of the task. Specifically, text descriptions (see Appendix I) are fed to "Phi-3 Mini-128k Instruct" (Microsoft, 2024) and we compute the channel-wise mean over the features of penultimate layer as a 50 dimensional parameterization for each task. We use $K = 3$ and $\epsilon = .7$.

Additionally, we highlight that we use success rate instead of returns as our evaluation metric in Meta-World, as it has been the standard metric across different papers.

### 4.1. Baselines and Evaluation

We compare our approaches to 11 state-of-the-art baselines. Five of these are designed for MTRL: 1) CMTA (Lan et al., 2024), 2) MOORE (Hendawy et al., 2024), 3) CARE (Sodhani et al., 2021), 4) soft modularization (Soft) (Yang et al., 2020), and 5) Multi task SAC (Yu et al., 2020a). Five more are meta-RL algorithms: 1) MAML (Finn et al., 2017a), 2) RL2 (Duan et al., 2017), 3) PEARL (Rakelly et al., 2019), 4) VariBAD (Zintgraf et al., 2020), and 5) AMAGO (Grigsby et al., 2024). Finally, we also compare to the state-of-the-art approach using expectation–maximization (EM) to learn a policy committee (EM) (Ivanov & Ben-Porat, 2024).

Our evaluation involves three settings: *training*, *test*, and *few-shot*. The training evaluation corresponds to standard MTRL. The test evaluation uses a test set to evaluate all approaches with no fine-tuning. Finally, our few-shot test evaluation allows a short round of fine-tuning on the test data. For PACMAN we select the best-performing policy for training and test, and use the proposed few-shot approach to *learn* the best policy through empirical policy evaluation for the few-shot test setting (see Section 3.4). In all figures, error bounds are 1 sample standard deviation.

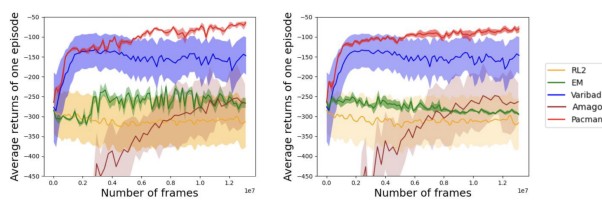

*Figure 2.* HalfCheetahVel train (left) and test (right) comparisons.

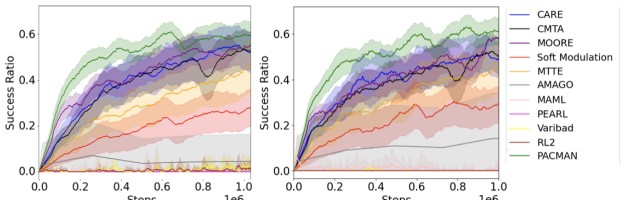

*Figure 3.* MetaWorld train (left) and test (right) comparisons.

## 4.2. Results

**MuJoCo** In the MuJoCo environment, we focus on *personalization*, varying only reward functions and focusing on the ability to generalize to a diverse set of rewards. Consequently, our baselines here include meta-RL approaches (RL2, VariBAD, AMAGO) and EM (personalized RL, which requires the dynamics to be shared across tasks, therefore not applicable for Meta-World), and PACMAN uses VariBAD as the within-cluster RL method.

Figure 2 presents the MuJoCo results for the training and test evaluations in HalfCheetahVel. We can see that PACMAN consistently outperforms the baselines in both evaluations, with VariBAD the only competitive baseline. The advantage of PACMAN is also pronounced in HumanoidDir, whose results are deferred to Appendix H.

*Table 1.* Few-shot learning effectiveness (MuJoCo).

|         | **Halfcheetah**    | **Humanoid**       |
| ------- | ------------------ | ------------------ |
| RL2     | -314.37 $\pm$ 1.15 | 946.17 $\pm$ 0.73  |
| VariBAD | -137.99 $\pm$ 1.14 | 1706.38 $\pm$ 0.75 |
| EM      | -325.29 $\pm$ 1.84 | 947.06 $\pm$ 0.84  |
| Amago   | -279.13 $\pm$ 0.67 | 1533.45 $\pm$ 0.49 |
| **PACMAN** | **-54.03 $\pm$ 1.34** | **2086.50 $\pm$ 0.89** |

The few-shot comparison is provided in Table 1, where the advantage of PACMAN is especially notable. In HalfcheetahVel, the improvement over the best baseline is by a factor of more than 2.5, while in HumanoidDir it is over 22%.

**Meta-World** Next, we turn to the complex multi-task Meta-World environment. In this environment, our approach uses MOORE for within-cluster training. Figure 3 presents the results for training and test evaluations, where we compare to the MTRL baselines (all meta-RL baselines are significantly worse on these metrics, likely because the goals of these algorithms are primarily efficacy in few-shot settings).

We observe that PACMAN significantly outperforms all baselines in both train and test cases (e.g., ~25% improvement over the best baseline after 500K steps).

The results for few-shot learning are provided in Table 2. Performance is a moving average success rate for the last

*Table 2.* Few-Shot Learning Results.

| Method | 6K Updates | 12K Updates |
| --- | --- | --- |
| MAML | 0.0025 $\pm$ 0.006 | 0.01 $\pm$ 0.03 |
| PEARL | 0.03 $\pm$ 0.03 | 0.27 $\pm$ 0.07 |
| RL2 | 0.007 $\pm$ 0.01 | 0.02 $\pm$ 0.02 |
| VariBAD | 0.025 $\pm$ 0.06 | 0.027 $\pm$ 0.07 |
| AMAGO | 0.08 $\pm$ 0.09 | .093 $\pm$ 0.09 |
| Soft | 0.27 $\pm$ 0.07 | 0.26 $\pm$ 0.08 |
| MTTE | 0.37 $\pm$ 0.08 | 0.40 $\pm$ 0.10 |
| CARE | 0.39 $\pm$ 0.05 | 0.40 $\pm$ 0.06 |
| CMTA | 0.45 $\pm$ 0.07 | 0.34 $\pm$ 0.08 |
| MOORE | 0.41 $\pm$ 0.08 | 0.44 $\pm$ 0.11 |
| **PACMAN** | **0.53 $\pm$ 0.02** | **0.60 $\pm$ 0.02** |

2000 evaluation episodes over 3 seeds. Here, the advantage of PACMAN over all baselines is particularly notable. First, somewhat surprisingly, the meta-RL baselines, with the exception of PEARL, underperform MTRL baselines in this setting. This is because our evaluation is significantly more challenging, with only 30 training tasks but with 20 diverse test tasks, and the adaptation phase has a very short (6-12K updates) time horizon for few-shot training, than typical in prior work. In contrast, MTRL methods fare reasonably well. The proposed PACMAN approach, however, significantly outperforms all the baselines. For example, only 12K updates suffice to reliably identify the best policy (comparing with zero-shot results in Table 2), with the result outperforming the best baseline by >36%.

### 4.3. Further Empirical Investigation of Our Algorithm

We investigate our algorithmic contribution in two ways. First, we compare our method with three common clustering methods in Meta-World's zero-shot setting: KMeans++ (Arthur & Vassilvitskii, 2006), DBScan (Khan et al., 2014), GMM (Bishop, 2006), as well as with random clustering.

| Method | 125K Steps | 250K Steps |
| --- | --- | --- |
| KMeans++ | 0.22 $\pm$ 0.06 | 0.30 $\pm$ 0.07 |
| DBScan | 0.19 $\pm$ 0.04 | 0.28 $\pm$ 0.06 |
| GMM | 0.29 $\pm$ 0.07 | 0.33 $\pm$ 0.08 |
| Random | 0.22 $\pm$ 0.04 | 0.25 $\pm$ 0.04 |
| **PACMAN (Ours)** | **0.36 $\pm$ 0.07** | **0.48 $\pm$ 0.09** |

*Table 3.* Performance Comparison for different clustering methods at 125K and 250K Steps.

Notably, PACMAN exhibits $\sim 45\%$ performance improve-

ment over the next best clustering method for $K = 3$. We have also conducted the same ablations over the MuJoCo setting, where the advantage of our method is also significant; see Table 4 in Appendix H.2 for more details. In addition, Figure 5 in Appendix H.2 compares a histogram of rewards for PACMAN and KMeans++ in the MuJoCo Halfcheetah environment, showing a consistent distributional improvement (not merely in expectation), excepting a small number of outliers.

Additionally, we consider the impact of varying budget $K$. We observe that efficacy of PACMAN is non-monotonic in $K$. The reason is that once $K$ is sufficiently large to cover the entire set of training tasks, increasing it further reduces the number of training tasks in individual clusters and thereby hurts generalization, both within clusters and to previously unseen tasks. Finally, we also consider the impact of changing the hyperparameter $\epsilon$, and observe that the results are relatively robust to small changes in $\epsilon$.

Overall, our method stands out by introducing only a single hyperparameter, $\epsilon$, which proves easy to tune. This simplicity is a significant advantage, especially when contrasted with the myriad of hyperparameters often encountered in typical deep RL methods. A practical approach to setting $\epsilon$ is to first compute the distances between all task embeddings and subsequently fine-tune it to ensure it remains small while still providing adequate coverage. Notably, our ablation studies further indicate that the performance of our method is far less sensitive to $\epsilon$ compared to the typical hyperparameter sensitivity observed in RL. See Appendix H for further details on these ablations.

## 5. Conclusion and Limitations

We developed a general algorithmic framework for learning policy committees for effective generalization and few-shot learning in multi-task settings with diverse tasks that may be unknown at training time. We showed that our approach is theoretically grounded, and outperforms MTRL, meta-RL, and personalized RL baselines in both training, and zero-shot and few-shot test evaluations, often by a large margin. Nevertheless, our approach exhibits several important limitations. First, it requires tasks to be parametric, and while we demonstrate how LLMs can be used to effectively obtain task embeddings in the Meta-World environments, it is not clear how to do so generally. Second, it includes a scalar hyperparameter, $\epsilon$, which determines how we evaluate the quality of task coverage and needs to be adjusted separately for each environment, although this hyperparameter is easily tunable in practice.

## Acknowledgements

This work was supported by the National Science Foundation (NSF) under grants IIS-2214141 and CCF-2403758, the Office of Naval Research (ONR) under award N00014-24-1-2663, the Army Research Office (ARO) under grant W911NF-25-1-0059, and with the generous support of NVIDIA Corporation.

## Impact Statement

This paper provides a theoretically grounded and empirically validated framework for personalization in Markov Decision Processes (MDPs) through the construction of policy committees.The simplicity and modularity of our approach enable wide applicability across domains such as robotics, adaptive systems, and user-centered decision-making frameworks. Furthermore, our method inherently promotes fairness by ensuring that a diverse range of tasks are represented by at least one effective policy in the committee. The clustering strategy may also extend beyond reinforcement learning to other domains where balancing efficiency and individualization is critical. By offering a scalable and provably efficient solution, our work lays a foundation for future advances in equitable and adaptive AI systems.

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

## A. Proof of Corollary 2.2

**Definition A.1.** A *potential function* is a function $\Phi : \mathcal{S} \to \mathbb{R}$. Given a discount $\gamma$, $r_1$ and $r_2$ differ by potential shaping if for some potential $\Phi$, we have that $r_2(s, a, s') = r_1(s, a, s') + \gamma \cdot \Phi(s') - \Phi(s)$.

**Definition A.2.** Given a transition function $\mathcal{T}$, $r_1$ and $r_2$ differ by $S'$-*redistribution* if $\mathbb{E}_{S' \sim \mathcal{T}(s,a)}[r_2(s, a, s')] = \mathbb{E}_{S' \sim \mathcal{T}(s,a)}[r_1(s, a, s')]$.

**Definition A.3.** $r_1$ and $r_2$ differ by positive linear scaling if $r_2(s, a, s') = c \cdot r_1(s, a, s')$ for some positive constant $c$.

*Proof.* Theorem 2.6 from (Skalse & Abate, 2023) says that two tasks $\tau_1, \tau_2$ have the same ordering of policies if and only if $r_1$ and $r_2$ differ by potential shaping, positive linear scaling, and $S'$-redistribution. Therefore, if we could find optimal policies for these two tasks separately; they necessarily differ. And by forming a policy committee of these two optimal policies, we have $V_{\tau_1}^\Pi \geq V_{\tau_1}^\pi$, $V_{\tau_2}^\Pi \geq V_{\tau_2}^\pi$, and $V_{\tau_1}^\Pi + V_{\tau_2}^\Pi > V_{\tau_1}^\pi + V_{\tau_2}^\pi$ for any policy $\pi$. □

## B. Proof of Theorem 3.3

**Definition B.1** (Gap preserving reduction for a maximization problem). Assume $\Pi_1$ and $\Pi_2$ are some maximization problems. A gap-preserving reduction from $\Pi_1$ to $\Pi_2$ comes with four parameters (functions) $f_1, \alpha, f_2$ and $\beta$. Given an instance $x$ of $\Pi_1$, the reduction computes in polynomial time an instance $y$ of $\Pi_2$ such that: $OPT(x) \geq f_1(x) \implies OPT(y) \geq f_2(y)$ and $OPT(x) < \alpha |x| f_1(x) \implies OPT(y) < \beta |y| f_2(y)$.

*Proof.* Let $G = (V, E)$ be an undirected graph with 5 vertices and 2 edges as follows:

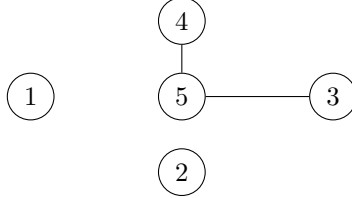

We create an instance of Max-coverage for a set of $\theta$s in $\mathbb{R}^n$ by filling out their coordinate matrix $A_{ij} = \begin{cases} 0 & \text{if } i = j \\ 1.5\epsilon & \text{if } i, j \text{ are adjacent} \\ 2.5\epsilon & \text{if } i, j \text{ are not adjacent} \end{cases}$ :

| dim | $\theta_1$ | $\theta_2$ | $\theta_3$ | $\theta_4$ | $\theta_5$ |
|---|---|---|---|---|---|
| 1 | 0 | 2.5 | 2.5 | 2.5 | 2.5 |
| 2 | 2.5 | 0 | 2.5 | 2.5 | 2.5 |
| 3 | 2.5 | 2.5 | 0 | 2.5 | 1.5 |
| 4 | 2.5 | 2.5 | 2.5 | 0 | 1.5 |
| 5 | 2.5 | 2.5 | 1.5 | 1.5 | 0 |

Let $\theta_1 = [0, 2.5, 2.5, 2.5, 2.5]$, $\theta_2 = [2.5, 0, 2.5, 2.5, 2.5]$, $\theta_3 = [2.5, 2.5, 0, 2.5, 1.5]$, $\theta_4 = [2.5, 2.5, 2.5, 0, 2.5]$, $\theta_5 = [2.5, 2.5, 1.5, 1.5, 0]$.

Projected onto the fifth axis, our thetas look like:

$$x_5 - \epsilon \; \longrightarrow\!\!\times\!\!\longrightarrow \; x_5 + \epsilon$$
$$x_1 - \epsilon \; \longrightarrow\!\!\times\!\!\longrightarrow \; x_1 + \epsilon$$
$$x_2 - \epsilon \; \longrightarrow\!\!\times\!\!\longrightarrow \; x_2 + \epsilon$$
$$x_3 - \epsilon \; \longrightarrow\!\!\times\!\!\longrightarrow \; x_3 + \epsilon$$
$$x_4 - \epsilon \; \longrightarrow\!\!\times\!\!\longrightarrow \; x_4 + \epsilon$$

And similarly, onto the third axis:

$$x_3 - \epsilon \;\longrightarrow\!\!\times\!\!\longrightarrow\; x_3 + \epsilon$$
$$x_1 - \epsilon \;\longrightarrow\!\!\times\!\!\longrightarrow\; x_1 + \epsilon$$
$$x_2 - \epsilon \;\longrightarrow\!\!\times\!\!\longrightarrow\; x_2 + \epsilon$$
$$x_4 - \epsilon \;\longrightarrow\!\!\times\!\!\longrightarrow\; x_4 + \epsilon$$
$$x_5 - \epsilon \;\longrightarrow\!\!\times\!\!\longrightarrow\; x_5 + \epsilon$$

We claim that we have constructed a gap-preserving reduction for any $t > 0$

$$OPT(A) = n \implies OPT(B) = n$$

$$OPT(A) < n^{1-t} \implies OPT(B) < n^{1-t}.$$

To begin with, if the Max-Clique instance consists of a complete graph, then the $\theta$s we created have coordinates equal to $1.5\epsilon$ everywhere except $i$-th coordinate, which is zero. So they can all be covered by one $\tilde{\theta} = [0.7\epsilon, 0.7\epsilon, \ldots, 0.7\epsilon]$, the coverage size is $n$. Therefore, the first implication is true.

Then for the second statement, we argue with the contrapositive: assume that one of the maximum coverage sets is $S = \{i_1, \ldots, i_k\}$ and $k \geq n^{1-t}$. We have to prove that the maximum clique has size greater than or equal to $k \geq n^{1-t}$.

Specifically, we prove that the vertices corresponding to the elements from $S$ form a clique.

If $\theta_i, \theta_j$ are from the set $S$, then they should be covered on each dimension since the $||\theta_i - \theta_j||_\infty = \max |\theta_i^d - \theta_j^d| \leq \epsilon$. So $\theta_i, \theta_j$ have to be adjacent, because otherwise their corresponding coordinates on the $i$-th and $j$-th dimension are more than $\epsilon$ away. For example, we have theta $\theta_3^3 = 0$ and $\theta_5^5 = 0$, so $\theta_5^3$ and $\theta_5^3$ must be $1.5\epsilon$ rather than $2.5\epsilon$, which indicates that $3, 5$ are neighbors in the graph.

Therefore, the points in $S$ correspond to a clique of size $k \geq n^{1-t}$ in the graph. Thus, if the graph $G$ has a clique of size less than $n^{1-t}$, then the maximum coverage set has size less than $n^{1-t}$. $\qquad\square$

## C. Pseudocode of Greedy Intersection Algorithm

The full pseudocode for the *Greedy Intersection Algorithm (GIA)* algorithm is provided as Algorithm 1.

## D. Proof of Theorem 3.4

Based on the proof of maxmizing monotone submodular functions by (Nemhauser et al., 1978).

**Lemma D.1.** *Suppose $1 - \delta^*(K)$ is the optimal $(\epsilon, 1 - \delta)$-parameter-cover of $\Gamma$ achievable with fixed $K$. With probability at least $1 - \alpha$, the probability of $\theta$ from $\Gamma$ getting covered by the first $i$ representatives generated by Algorithm 1 is greater than $\frac{1 - \delta^*(K) - K\beta}{K} \sum_{j=0}^{i-1}(1 - 1/K)^j$.*

*Proof.* We will prove the lemma through induction. We begin by defining the coverage region of each of the $K$ committee member in the optimal parameter-cover as $S_i^*$. Furthermore, let $\Pi^*$ denote the region covered by this optimal parameter-cover. Thus, $\Pi^* = \bigcup S_i^*$. Next, let $A_i$ denote the region covered by the representative selected on the $i$-th iteration. And let $C_i$ denote the set of $\theta$s from the dataset $T$ that are covered after $i$-th iteration.

First of all, we want to show at $i = 1$, the probability for $\theta \sim \Gamma$ getting covered is greater than $\frac{1 - \delta^*(K) - K\beta}{K} \sum_{j=0}^{0}(1 - 1/K)^0 = \frac{1 - \delta^*(K) - K\beta}{K}$.

By Hoeffding's theorem, $\Pr_{\theta \sim \Gamma}[\mathbb{E}_{\theta \sim \Gamma}[\mathbf{1}(\theta \in \bigcup_{j=1}^{i-1} A_j)] - \frac{\sum_i \mathbf{1}(\theta_i \in \bigcup_{j=1}^{i-1} A_j)}{N}) \geq \frac{\beta}{3}] \leq \exp(-2N\beta^2/9) = \frac{\alpha}{5}$. Hence, with probability at least $1 - \frac{\alpha}{5}$, $\Pr_{\theta \sim \Gamma}[\theta \in \bigcup_{j=1}^{i-1} A_j] = \mathbb{E}_{\theta \sim \Gamma}[\mathbf{1}(\theta \in \bigcup_{j=1}^{i-1} A_j)] \leq \frac{\sum_i \mathbf{1}(\theta_i \in \bigcup_{j=1}^{i-1} A_j)}{N} + \frac{\beta}{3} = \frac{|C_{i-1}|}{N} + \frac{\beta}{3}$.

Now the union bound first gives that $\Pr_{\theta \sim \Gamma}[\theta \in \Pi^* \wedge \theta \notin \bigcup_{j=1}^{i-1} A_j] \geq \Pr_{\theta \sim \Gamma}[\theta \in \Pi^*] - \Pr_{\theta \sim \Gamma}[\theta \in \bigcup_{j=1}^{i-1} A_j] = 1 - \delta^*(K) - \Pr_{\theta \sim \Gamma}[\theta \in \bigcup_{j=1}^{i-1} A_j]$. Applying union bound again, we obtain that with probability at least $1 - \alpha_1$, $\sum_{i=1}^{K} \Pr_{\theta \sim \Gamma}[\theta \in S_i^* \wedge \theta \notin \bigcup_{j=1}^{i-1} A_j] \geq \Pr_{\theta \sim \Gamma}[\theta \in \Pi^* \wedge \theta \notin \bigcup_{j=1}^{i-1} A_j] \geq 1 - \delta^*(K) - (\frac{|C_{i-1}|}{N} + \frac{\beta}{3})$. Hence, $\max_{i \in [K]} \Pr_{\theta \sim \Gamma}[\theta \in S_i^* \wedge \theta \notin \bigcup_{j=1}^{i-1} A_j] \geq \frac{1 - \delta^*(K) - (\frac{|C_{i-1}|}{N} + \frac{\beta}{3})}{K}$. Let us call this maximising $S_i^* \; \hat{S}$.

---

**Algorithm 1** Greedy Intersection

---

**Input**: $T = \{\theta_i\}_{i=1}^{N}$, $\epsilon > 0$, $K \geq 1$
**Output**: Parameter cover $C$

1:   $C \leftarrow []$
2: **for** $round\ k = 1$ to $K$ **do**
3:     **for** $dimension\ m = 1$ to $d$ **do**
4:        Sort $T$ in ascending order based on their $m$-th coordinates
5:        $lists_m \leftarrow []$
6:        **for** $indiviual\ i = 2$ to $N$ **do**
7:          $S_i \leftarrow [\theta_i]$
8:          **for** $j = i - 1$ to $1$ **do**
9:            **if** $\theta_i$'s $m$-th coordinate $< \theta_j$'s $m$-th coordinate $+2\epsilon$ **then**
10:              Add $\theta_j$ to $S_i$
11:            **else**
12:              **if** $lists_m[-1] \subseteq S_i$ **then**
13:                 $lists_m[-1] \leftarrow S_i$
14:              **else**
15:                 Add $S_i$ to $lists_m$
16:              **end if**
17:              **break**
18:            **end if**
19:          **end for**
20:        **end for**
21:     **end for**
22:     $S^{1*}, \ldots, S^{m*} \leftarrow \operatorname{argmax}_{S^1 \in lists_1, \ldots, S^m \in lists_m} |S^1 \cap \cdots \cap S^m|$
23:     $covered \leftarrow S^{1*} \cap \cdots \cap S^{m*}$
24:     $\hat{\theta}_k \leftarrow$ average of the $covered$
25:     $T \leftarrow T - covered$
26:     $C$.adds($\hat{\theta}_k$)
27: **end for**
28: **return** $C$

---

According to our Algorithm 1, $A_i$ covers the most $\theta$s from $T$ that were not covered in the previous rounds by $\bigcup_{j=1}^{i-1} A_j$. In particular, $|C_i| - |C_{i-1}|$ is greater or equal to the number of $\theta$s from $T$ covered in $\hat{S}$ but not $\bigcup_{j=1}^{i-1} A_j$. Let us denote the latter as $s_1$, and the former as $s_2$, then $s_1 - s_2 \leq 0$.

Hoeffding's theorem gives us $\Pr_{\theta \sim \Gamma}(\mathbb{E}_{\theta \sim \Gamma}[\mathbf{1}[\theta \in \hat{S} \wedge \theta \notin \bigcup_{j=1}^{i-1} A_j]] - s_1/N) \geq \frac{\beta}{6}) \leq (\frac{\alpha}{5})^4$ and $\Pr_{\theta \sim \Gamma}(s_2/N - \mathbb{E}_{\theta \sim \Gamma}[\mathbf{1}[\theta \in A_i \wedge \theta \notin \bigcup_{j=1}^{i-1} A_j]] \geq \frac{\beta}{6}) \leq (\frac{\alpha}{5})^4$. Hence with probability at least $1 - 2(\frac{\alpha}{5})^4$, $\mathbb{E}_{\theta \sim \Gamma}[\mathbf{1}[\theta \in \hat{S} \wedge \theta \notin \bigcup_{j=1}^{i-1} A_j]] - \mathbb{E}_{\theta \sim \Gamma}[\mathbf{1}[\theta \in A_i \wedge \theta \notin \bigcup_{j=1}^{i-1} A_j]] = (\mathbb{E}_{\theta \sim \Gamma}[\mathbf{1}[\theta \in \hat{S} \wedge \theta \notin \bigcup_{j=1}^{i-1} A_j]] - s_1/N) + (s_1 - s_2)/N + (s_2/N - \mathbb{E}_{\theta \sim \Gamma}[\mathbf{1}[\theta \in A_i \wedge \theta \notin \bigcup_{j=1}^{i-1} A_j]] \leq \frac{\beta}{6} + \frac{\beta}{6} = \frac{\beta}{3}$.

Applying the result we obtained at the beginning of the proof, we have with probability at least $1 - \frac{\alpha}{5} - 2(\frac{\alpha}{5})^4$,

$$\Pr_{\theta \sim \Gamma}[\theta \in A_i \wedge \theta \notin \bigcup_{j=1}^{i-1} A_j]$$

$$\geq \Pr_{\theta \sim \Gamma}[\theta \in \hat{S} \wedge \theta \notin \bigcup_{j=1}^{i-1} A_j] - \frac{\beta}{3}$$

$$\geq \frac{1 - \delta^*(K) - (\frac{|C_{i-1}|}{N} + \frac{\beta}{3})}{K} - \frac{\beta}{3}. \tag{3}$$

Since nothing is covered before the first iteration, we can use step (3) with $|C_0| = 0$ to prove the base condition for the claim. Because $K \geq 1$, we have $\frac{1 - \delta^*(K) - \frac{\beta}{3}}{K} - \frac{\beta}{3} = \frac{1 - \delta^*(K) - \frac{(1+K)\beta/3}{K}}{K} \geq \frac{1 - \delta^*(K) - K\beta}{K}$.

The induction hypothesis is that for all $i \leq K - 1$, we have $\Pr_{\theta \sim \Gamma}[\theta \in \bigcup_{j=1}^{i} A_j] \geq \frac{1 - \delta^*(K) - K\beta}{K} \sum_{j=0}^{i} (1 - 1/K)^j$.

By Hoeffding, $\Pr_{\theta \sim \Gamma}[|\Pr_{\theta \sim \Gamma}[\theta \in \bigcup_{j=1}^{i-1} A_j] - \frac{|C_{i-1}|}{N}| \geq \beta/3] \leq 2\exp(-2N\beta^2/9)$. In other words, with probability at least $1 - 2\frac{\alpha}{5}$, $\Pr_{\theta \sim \Gamma}[\theta \in \bigcup_{j=1}^{i-1} A_j] \geq \frac{|C_{i-1}|}{N} - \beta/3$ and $\frac{|C_{i-1}|}{N} \geq \Pr_{\theta \sim \Gamma}[\theta \in \bigcup_{j=1}^{i-1} A_j] - \beta/3$.

Then at the step $i = K$, since for $\frac{\alpha}{5} \in (0, 1), (\frac{\alpha}{5})^4 < \frac{\alpha}{5}$, we have with probability at least $1 - 2\frac{\alpha}{5} - \frac{\alpha}{5} - 2(\frac{\alpha}{5})^4 \geq 1 - 5\frac{\alpha}{5} = 1 - \alpha$,

$$
\Pr_{\theta \sim \Gamma}[\theta \in \bigcup_{j=1}^{i} A_j]
$$

$$
= \Pr_{\theta \sim \Gamma}[\theta \in \bigcup_{j=1}^{i-1} A_j] + \Pr_{\theta \sim \Gamma}[\theta \in A_i \wedge \theta \notin \bigcup_{j=1}^{i-1} A_j]
$$

$$
\geq \frac{|C_{i-1}|}{N} - \frac{\beta}{3} + \frac{1 - \delta^*(K) - (\frac{|C_{i-1}|}{N} + \beta/3)}{K} - \frac{\beta}{3}
$$

$$
= \frac{1 - \delta^*(K)}{K} + (1 - 1/K)\frac{|C_{i-1}|}{N} - \frac{(2K+1)\beta}{3K}
$$

$$
\geq \frac{1 - \delta^*(K)}{K} + (1 - 1/K)(\Pr_{\theta \sim \Gamma}[\theta \in \bigcup_{j=1}^{i-1} A_j] - \beta/3) - \frac{(2K+1)\beta}{3K}
$$

$$
\geq \frac{1 - \delta^*(K)}{K} + (1 - 1/K)\left(\frac{1 - \delta^*(K) - K\beta}{K}\sum_{j=0}^{i-1}(1 - 1/K)^j\right) - (1 - 1/K)\beta/3 - \frac{(2K+1)\beta}{3K}
$$

$$
= \frac{1 - \delta^*(K)}{K} - \frac{(2K + 1 + K - 1)\beta}{3K} + \frac{1 - \delta^*(K) - K\beta}{K}\sum_{j=1}^{i}(1 - 1/K)^j
$$

$$
= \frac{1 - \delta^*(K) - K\beta}{K}\sum_{j=0}^{i}(1 - 1/K)^j
$$

$\square$

*Proof of Theorem 3.4.* We can directly apply lemma D.1 to $i = K$. Call the region defined by the cover generated by Algorithm 1 $\Pi_K = \bigcup_{j=1}^{K} A_j$. Using the inequality $(1 - 1/K)^K \geq 1 - 1/e$ for all $K \geq 0$, we have

$$
\Pr_{\theta \sim \Gamma}[\theta \in \Pi_K] \geq \frac{1 - \delta^*(K) - K\beta}{K}\sum_{j=0}^{K}(1 - 1/K)^j
$$

$$
= \frac{1 - \delta^*(K) - K\beta}{K}\frac{1 - (1 - 1/K)^K}{1 - (1 - 1/K)}
$$

$$
= (1 - \delta^*(K) - K\beta)(1 - (1 - 1/K)^K)
$$

$$
\geq (1 - 1/e)(1 - \delta^*(K) - K\beta).
$$

$\square$

# E. Proof of Theorem 3.5

*Proof.* Let us call the optimal solutions set to (1) $A_1$, and the optimal solutions set to (2) $A_2$.

We first show $A_1 \subset A_2$. Pick any $\{\theta_1, \ldots, \theta_K\} \in A_1$. Due to the premise, for each $i$, since $\min_{k \in [K]} \|\theta_k - \theta_i\|_\infty - \epsilon \leq 0$, there exists $\theta_{k^*}$ such that $\|\theta_{k^*} - \theta_i\|_\infty - \epsilon \leq 0$. Thus, we can have $w_{ik^*} = 1$, and $w_{ik} = 0$ for all the other $k \neq k^*$. Then $\mathbf{ReLU}\left(\left\{\sum_{k \in [K]} \sigma(w_{ik})\|\theta_k - \theta_i\|_\infty\right\} - \epsilon\right) = \mathbf{ReLU}(\|\theta_{k^*} - \theta_i\|_\infty - \epsilon) = 0$. By setting $w$ this way, we could achieve the zero loss for the relaxation problem. Hence $\{\theta_1, \ldots, \theta_K\} \in A_2$.

Now to show $A_2 \subset A_1$, suppose $\{\theta_1, \ldots, \theta_K\}, w$ is a optimal solution. Due to the premise, we must have that $\mathbf{ReLU}\left(\left\{\sum_{k \in [K]} \sigma(w_{ik})\|\theta_k - \theta_i\|_\infty\right\} - \epsilon\right) = 0$ for each $i$. Now fix $i$, since each $\sigma(w_{ik})$ is nonegative and summing them over $k$ yields 1, there must be some positive coordinate $\sigma(w_{ik'})$. Hence for all such $k'$, $\mathbf{ReLU}(\|\theta_{k'} - \theta_i\|_\infty - \epsilon) = 0$, i.e., $\|\theta_{k'} - \theta_i\|_\infty \leq \epsilon$. Thus, $\min_{k \in [K]} \|\theta_k - \theta_i\|_\infty \leq \|\theta_{k'} - \theta_i\|_\infty \leq \epsilon$ also holds, and $\max_{\{\theta_1, \ldots, \theta_K\}} \sum_i \mathbf{1}(\min_{k \in [K]} \|\theta_k - \theta_i\|_\infty \leq \epsilon) = n$. Consequently, $\{\theta_1, \ldots, \theta_K\} \in A_1$. $\qquad\square$

## F. Proof of Lemma 3.6

$$
\begin{aligned}
V_i^{\pi_i^*} &= \mathbb{E}[\sum_{t=0}^{T} \gamma^t r_{\theta_i}(s_t, a_t) \,|\pi_i^*] \\
&= \mathbb{E}[\sum_{t=0}^{T} \gamma^t (r_{\theta_i}(s_t, a_t) - r_{\theta_j}(s_t, a_t) + r_{\theta_j}(s_t, a_t)) \,|\pi_i^*] \\
&= \mathbb{E}[\sum_{t=0}^{T} \gamma^t (r_{\theta_i}(s_t, a_t) - r_{\theta_j}(s_t, a_t)) \,|\pi_i^*] + \mathbb{E}[\sum_{t=0}^{T} \gamma^t r_{\theta_j}(s_t, a_t) \,|\pi_i^*] \\
&= \mathbb{E}[\sum_{t=0}^{T} \gamma^t (r_{\theta_i}(s_t, a_t) - r_{\theta_j}(s_t, a_t)) \,|\pi_i^*] + V_j^{\pi_i^*} \\
&\leq \sum_{t=0}^{T} \gamma^t L \|\theta_i - \theta_j\|_\infty + V_j^{\pi_j^*}(-V_i^{\pi_j^*} + V_i^{\pi_j^*}) \\
&\leq L \frac{\gamma^{T+1} - 1}{\gamma - 1} \epsilon + (V_2^{\pi_j^*} - V_i^{\pi_j^*}) + V_i^{\pi_j^*} \\
&= L \frac{\gamma^{T+1} - 1}{\gamma - 1} \epsilon + \mathbb{E}[\sum_{t=0}^{T} \gamma^t r_{\theta_i}(s_t, a_t) - r_{\theta_j}(s_t, a_t) \,|\pi_j^*] + V_i^{\pi_j^*} \\
&\leq 2L \frac{\gamma^{T+1} - 1}{\gamma - 1} \epsilon + V_i^{\pi_j^*}
\end{aligned}
$$

If the discount factor $\gamma = 1$, the argument is as follows:

$$
\begin{aligned}
V_i^{\pi_i^*} &= \mathbb{E}[\sum_{t=0}^{T} r_\theta(s_t, a_t) \,|\pi_i^*] \\
&= \mathbb{E}[\sum_{t=0}^{T} r_\theta(s_t, a_t) - r_{\theta_j}(s_t, a_t) + r_{\theta'}(s_t, a_t) \,|\pi_i^*] \\
&= \mathbb{E}[\sum_{t=0}^{T} r_\theta(s_t, a_t) - r_{\theta_j}(s_t, a_t) \,|\pi_i^*] + \mathbb{E}[\sum_{t=0}^{T} (r_{\theta_j}(s_t, a_t) \,|\pi_i^*] \\
&= \mathbb{E}[\sum_{t=0}^{T} r_{\theta_i}(s_t, a_t) - r_{\theta_j}(s_t, a_t) \,|\pi_i^*] + V_j^{\pi_i^*} \\
&\leq \sum_{t=0}^{T} L \|\theta_i - \theta_j\|_\infty + V_j^{\pi_j^*}(-V_i^{\pi_j^*} + V_i^{\pi_j^*})
\end{aligned}
$$

$$\leq TL\epsilon + (V_j^{\pi_j^*} - V_i^{\pi_j^*}) + V_i^{\pi_j^*}$$

$$= TL\epsilon + \mathbb{E}[\sum_{t=0}^{T} r_{\theta_i}(s_t, a_t) - r_{\theta_j}(s_t, a_t) \mid \pi_j^*] + V_i^{\pi_j^*}$$

$$\leq 2TL\epsilon + V_i^{\pi_j^*}$$

# G. Proof of Theorem 3.8

We prove this by leveraging the following lemma by Azar et al. (2013).

**Definition G.1.** The average expected reward for a given policy is measured per time step as

$$\mu^\pi(s) = \frac{1}{h}\mathbb{E}\left[\sum_{t=1}^{h} r(s_t, \pi(s_t)) \mid s_0 = s\right].$$

And the empirical average return after $n$ episodes is

$$\hat{\mu}^\pi = \frac{1}{nh}\sum_{i=0}^{n}\sum_{t=0}^{h} r(s_t, \pi(s_t)).$$

**Assumption G.2.** There exists a policy $\pi^+ \in \Pi$, which induces a unichain Markov process on the MDP $M$, such that the average reward $\mu^{\pi^+} \geq \mu^\pi(s) \ \forall s \in \mathcal{S}$ and any policy $\pi \in \Pi$. The span of the bias function is $\mathrm{sp}(\lambda^{\pi^+}) = \max_s \lambda^{\pi^+}(s) - \min_s \lambda^{\pi^+}(s) \leq H$ for some constant $H$, where $\lambda$ the bias is defined as $\lambda^\pi(s) + \mu^\pi = \mathbb{E}[r(s, \pi(s)) + \lambda^\pi(s')]$, with $s'$ being the next state after the interaction $(s, \pi(s))$.

**Assumption G.3.** Suppose that each $\pi \in \Pi$ induces on the MDP $\mathcal{M}$ a single recurrent class with some additional transient states, i.e., $\mu^\pi(s) = \mu^\pi$ for all $s \in \mathcal{S}$, and $\mathrm{sp}(\lambda^\pi) \leq H$ for some finite $H$.

**Lemma G.4.** *(Azar et al., 2013, Lemma 1) Under Assumption G.2 and G.3, $|\hat{\mu}^\pi - \mu^\pi| \leq 2(H+1)\sqrt{\frac{2\log(2/\alpha)}{ph}} + \frac{H}{h}$ with probability at least $1 - \alpha$.*

*Proof.* Let $p = \frac{32h(H+1)^2 \log(4/\alpha)}{(\beta - 2H)^2}$. Denote the average rewards of the best and second best policy in the committee as $\mu^+, \mu^-$. If $\mu^+ - \mu^- > \beta/h$, by ensuring the difference between the estimation and the true average reward is small than $\beta/2h$. We can make sure we have picked the best policy. From Lemma G.4, we know $\Pr[\hat{\mu}^- \leq \mu^- + 2(H+1)\sqrt{\frac{2\log(4/\alpha)}{ph}} + \frac{H}{h}] = \Pr[\hat{\mu}^- \leq \mu^- + \beta/2h] \geq 1 - \alpha/2$. And $\Pr[\hat{\mu}^+ \geq \mu^+ - 2(H+1)\sqrt{\frac{2\log(4/\alpha)}{ph}} + \frac{H}{h}] = \Pr[\hat{\mu}^+ \leq \mu^+ - \beta/2h] \geq 1 - \alpha/2$. Hence with probability at least $1 - \alpha$, $\hat{\mu}^+ > \mu^+ - \beta/2h \geq \mu^- + \beta/h - \beta/2h = \mu^- + \beta/2h > \hat{\mu}^-$. Thus the empirically best policy we have picked is also the best in expectation. Now if $\mu^+ - \mu^- < \beta/h$, no matter which one we pick, we have the difference bound by $\beta/h$. The same holds for all pairs of policies ordered based on their expected values. Either way, with probability $1 - \alpha$, we could find the best policy in the committee. Since our committee is a $(\epsilon, 1 - \delta)$ cover, we are able to pick the policy with suboptimality $\beta + \epsilon$ with probability $1 - \delta - \alpha$. $\square$

# H. Additional Empirical Results

## H.1. Results on Humanoid Direction

We present the learning curves for both the training and zero-shot testing case as Figure H.1. The few shot result has been listed in Table 1.

## H.2. Additional results for empirical investigation of our method

### H.2.1. CLUSTERING ABLATIONS

We also obtained different clusters using PACMAN than using Kmeans++ and, as a result, much better performance for Halfcheetah-Velocity, as shown in the table below:

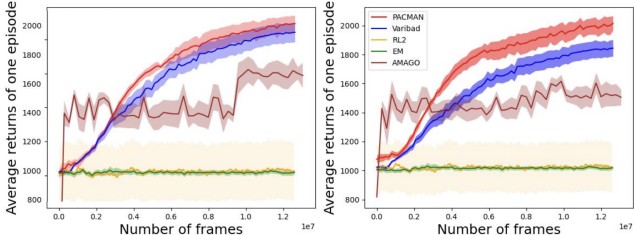

*Figure 4.* Humanoid-Direction Train and Zero Shot

*Table 4.* Comparison of clustering algorithms in HalfCheetah-Velocity, Zero-Shot.

| Method | 6 Million Frames | 12 Million Frames |
|---|---|---|
| KMeans++ | $-105.91 \pm 3.44$ | $-89.50 \pm 3.10$ |
| DBScan | $-234.05 \pm 9.56$ | $-213.42 \pm 4.09$ |
| GMM | $-239.32 \pm 11.27$ | $-199.86 \pm 9.54$ |
| Random | $-274.13 \pm 16.76$ | $-258.07 \pm 13.62$ |
| **PACMAN** | **-97.42 $\pm$ 3.70** | **-74.20 $\pm$ 6.76** |

As shown in Table **??**, the proposed approach outperforms all baselines.

While the performance of the KMeans++ algorithm appears relatively close to our method due to the significant gap between it and the other three clustering methods (DBSCAN, GMM, and Random), we emphasize that this result considers one hundred percent of the population.

The advantage of our algorithm becomes even more apparent when focusing on the welfare of the majority. To illustrate this, we present a histogram of rewards for individual test tasks during zero-shot testing using policies trained with our algorithm versus KMeans++ on the Half-Cheetah-Velocity benchmark:

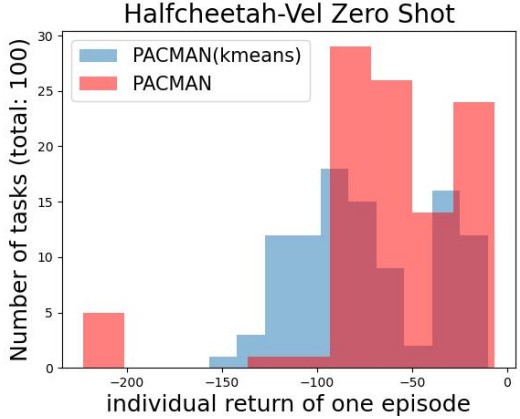

*Figure 5.* Histogram comparison of two clustering methods for zero-shot individual task rewards in Half-Cheetah (velocity).

The results vividly highlight a significantly greater density of high-performing tasks (red regions on the right) with our method. This suggests that our approach effectively promotes superior task performance while minimizing underperformance. In contrast, KMeans++ yields a more uniform but mediocre distribution of task performance. There is an ideological difference between these two clustering methods.

### H.2.2. HYPERPARAMETER ABLATIONS

We consider here additional ablations varying $K$ and $\epsilon$ omitted from the main body.

First, we present the results of ablations on $K$ on both Mujoco (Halfcheetah-Velocity) and Meta-World.

Both the ablation results for Meta-World and Mujoco demonstrate a clear advantage of utilizing a policy committee.

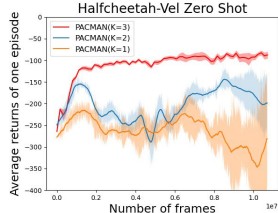 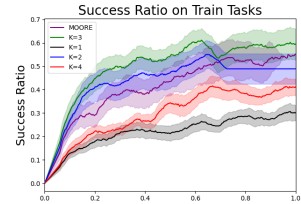 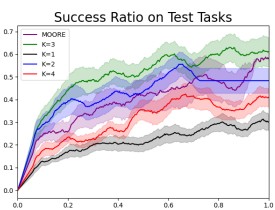

*Figure 6.* Varying $K$ from 1 to 3 for Halfcheetah-Velocity (left two) and 1 to 4 for Meta-World (right two)

*Table 5.* Few-shot in Meta-World, varying $K$.

| Method | 6K Updates | 12K Updates |
|---|---|---|
| MOORE | $0.42 \pm 0.06$ | $0.43 \pm 0.05$ |
| PACMAN ($K = 1$) | $0.32 \pm 0.05$ | $0.31 \pm 0.04$ |
| PACMAN ($K = 2$) | $0.50 \pm 0.05$ | $0.50 \pm 0.05$ |
| PACMAN ($K = 3$) | $0.61 \pm 0.04$ | $0.62 \pm 0.05$ |
| PACMAN ($K = 4$) | $0.32 \pm 0.05$ | $0.35 \pm 0.05$ |

Especially for Meta-World, our method beats the baseline for every $K$ greater than 1. Here, in few-shot settings, even using $K = 2$ already results in considerable improvement over the best baseline (MOORE), with $K = 3$ a significant further boost. Another thing to note is that increasing $K$ is not always better. The results in both the figure and the table show that as the number of tasks becomes increasingly partitioned, the generalization ability of each committee member may weaken. Hence the performance for $K = 4$ is worse than $K = 3$.

Finally, we show the effect of the $\epsilon$ hyperparameter in the Meta-World zero-shot setting. These results are reported as the success rate across all tasks for $K = 2$.

| | $\epsilon = .4$ | $\epsilon = .7$ | $\epsilon = 1$ |
|---|---|---|---|
| 500K Steps | 0.05 | 0.28 | 0.29 |
| 1M Steps | 0.05 | 0.31 | 0.40 |

*Table 6.* Success rate for $K = 2$.

We present also results for $K = 3$. The main results in the paper are for $\epsilon = .6$. Results are reported as the success rate over all tasks for 3 seeds.

| | $\epsilon = .5$ | $\epsilon = .6$ | $\epsilon = .7$ | $\epsilon = .8$ |
|---|---|---|---|---|
| 500K Steps | $0.23 \pm 0.08$ | $0.54 \pm 0.08$ | $0.56 \pm 0.09$ | $0.60 \pm 0.07$ |
| 1M Steps | $0.25 \pm 0.07$ | $0.60 \pm 0.10$ | $0.58 \pm 0.14$ | $0.58 \pm 0.07$ |

*Table 7.* Success rate for $K = 3$.

We find that increasing $\epsilon$ to cover more tasks can also improve performance (for a similar reason that increasing $K$ may not, as higher $\epsilon$ can ensure that we do not end up with clusters with too few tasks). Of course, for sufficiently high $\epsilon$, only a single cluster will emerge, so this, too induces an interesting tradeoff.

# I. Meta-World Task Embeddings

First, we manually generated the following task descriptions by referencing the metaworld documentation.

| Task Name | Objective | Environment Details |
|---|---|---|
| Reach-v1 | Move the robot's end-effector to a target position. | The task is set on a flat surface with random goal positions. The target position is marked by a small sphere or point in space. |
| Push-v1 | Push a puck to a specified goal position. | The puck starts in a random position on a flat surface. The goal position is marked on the surface. |
| Pick-Place-v1 | Pick up a puck and place it at a designated goal position. | The puck is placed randomly on the surface. The goal position is marked by a target area. |
| Door-Open-v1 | Open a door with a revolving joint. | The door can be opened by rotating it around the joint. Door positions are randomized. |
| Drawer-Open-v1 | Open a drawer by pulling it. | The drawer is initially closed and can slide out on rails. |
| Drawer-Close-v1 | Close an open drawer by pushing it. | The drawer starts in an open position. |
| Button-Press-Topdown-v1 | Press a button from the top. | The button is mounted on a panel or flat surface. |
| Peg-Insert-Side-v1 | Insert a peg into a hole from the side. | The peg and hole are aligned horizontally. |
| Window-Open-v1 | Slide a window open. | The window is set within a frame and can slide horizontally. |
| Window-Close-v1 | Slide a window closed. | The window starts in an open position. |
| Door-Close-v1 | Close a door with a revolving joint. | The door can be closed by rotating it around the joint. |
| Reach-Wall-v1 | Bypass a wall and reach a goal position. | The goal is positioned behind a wall. |
| Pick-Place-Wall-v1 | Pick a puck, bypass a wall, and place it at a goal position. | The puck and goal are positioned with a wall in between. |
| Push-Wall-v1 | Bypass a wall and push a puck to a goal position. | The puck and goal are positioned with a wall in between. |
| Button-Press-v1 | Press a button. | The button is mounted on a panel or surface. |
| Button-Press-Topdown-Wall-v1 | Bypass a wall and press a button from the top. | The button is positioned behind a wall on a panel. |
| Button-Press-Wall-v1 | Bypass a wall and press a button. | The button is positioned behind a wall. |
| Peg-Unplug-Side-v1 | Unplug a peg sideways. | The peg is inserted horizontally and needs to be unplugged. |
| Disassemble-v1 | Pick a nut out of a peg. | The nut is attached to a peg. |
| Hammer-v1 | Hammer a nail on the wall. | The robot must use a hammer to drive a nail into the wall. |
| Plate-Slide-v1 | Slide a plate from a cabinet. | The plate is located within a cabinet. |
| Plate-Slide-Side-v1 | Slide a plate from a cabinet sideways. | The plate is within a cabinet and must be removed sideways. |
| Plate-Slide-Back-v1 | Slide a plate into a cabinet. | The robot must place the plate back into a cabinet. |

| | | |
|---|---|---|
| Plate-Slide-Back-Side-v1 | Slide a plate into a cabinet sideways. | The plate is positioned for a sideways entry into the cabinet. |
| Handle-Press-v1 | Press a handle down. | The handle is positioned above the robot's end-effector. |
| Handle-Pull-v1 | Pull a handle up. | The handle is positioned above the robot's end-effector. |
| Handle-Press-Side-v1 | Press a handle down sideways. | The handle is positioned for sideways pressing. |
| Handle-Pull-Side-v1 | Pull a handle up sideways. | The handle is positioned for sideways pulling. |
| Stick-Push-v1 | Grasp a stick and push a box using the stick. | The stick and box are positioned randomly. |
| Stick-Pull-v1 | Grasp a stick and pull a box with the stick. | The stick and box are positioned randomly. |
| Basketball-v1 | Dunk the basketball into the basket. | The basketball and basket are positioned randomly. |
| Soccer-v1 | Kick a soccer ball into the goal. | The soccer ball and goal are positioned randomly. |
| Faucet-Open-v1 | Rotate the faucet counter-clockwise. | The faucet is positioned randomly. |
| Faucet-Close-v1 | Rotate the faucet clockwise. | The faucet is positioned randomly. |
| Coffee-Push-v1 | Push a mug under a coffee machine. | The mug and coffee machine are positioned randomly. |
| Coffee-Pull-v1 | Pull a mug from a coffee machine. | The mug and coffee machine are positioned randomly. |
| Coffee-Button-v1 | Push a button on the coffee machine. | The coffee machine's button is positioned randomly. |
| Sweep-v1 | Sweep a puck off the table. | The puck is positioned randomly on the table. |
| Sweep-Into-v1 | Sweep a puck into a hole. | The puck is positioned randomly on the table near a hole. |
| Pick-Out-Of-Hole-v1 | Pick up a puck from a hole. | The puck is positioned within a hole. |
| Assembly-v1 | Pick up a nut and place it onto a peg. | The nut and peg are positioned randomly. |
| Shelf-Place-v1 | Pick and place a puck onto a shelf. | The puck and shelf are positioned randomly. |
| Push-Back-v1 | Pull a puck to a goal. | The puck and goal are positioned randomly. |
| Lever-Pull-v1 | Pull a lever down 90 degrees. | The lever is positioned randomly. |
| Dial-Turn-v1 | Rotate a dial 180 degrees. | The dial is positioned randomly. |
| Bin-Picking-v1 | Grasp the puck from one bin and place it into another bin. | The puck and bins are positioned randomly. |
| Box-Close-v1 | Grasp the cover and close the box with it. | The box cover is positioned randomly. |
| Hand-Insert-v1 | Insert the gripper into a hole. | The hole is positioned randomly. |
| Door-Lock-v1 | Lock the door by rotating the lock clockwise. | The lock is positioned randomly. |
| Door-Unlock-v1 | Unlock the door by rotating the lock counter-clockwise. | The lock is positioned randomly. |

Our test tasks are the following: *assembly*, *basketball*, *bin picking*, *box close*, *button press topdown*, *button press topdown-wall*, *button press*, *button press wall*, *coffee button*, *coffee pull*, *coffee push*, *dial turn*, *disassemble*, *door close*, *door lock*,

*door open*, *door unlock*, *drawer close*, *drawer open* , and *faucet close*.

Then for each task, we fed the following prompt to an LLM:

```
"<Task Name>": "Objective: <Objective description>
```

```
Environment Details: <Environment details>"
```

For example:

```
"Reach-v1": "Objective: Move the robot's end-effector to a target position.
Environment Details: The task is set on a flat surface with random goal positions.
The target position is marked by a small sphere or point in space."
```

Next, we extracted the penultimate layer's activations of the LLM and computed the channel-wise mean. The result was a single 50-dimensional vector that represented the task.

## J. Meta-World Clustering Analysis and Discussion

Simply put, our method works by having committee members which are innately specialized to specific tasks, as illustrated below. Here committee member 2 is specialized to *door open* and committee member 3 is specialized to *door close*. At the same time, committee member 2 performs *door close* poorly and committee member 2 performs *door open* poorly. A MTRL policy in trying to perform all tasks doesn't perform any particular task well. Our method will select committee member 2 for *door open* and committee member 3 for *door close*.

To understand if the parametrization discussed in section 3.4 produces suitable clusters we have applied PCA to PCA to a clustering of 10 tasks. We note that the window tasks and drawer tasks are close in task space. Additionally, the dynamics and goals of the *push* and *pick-place* tasks are nearly identical. *Window close* is close to *door open* as both these tasks have the agent needing to move to the horizontally to begin the task.

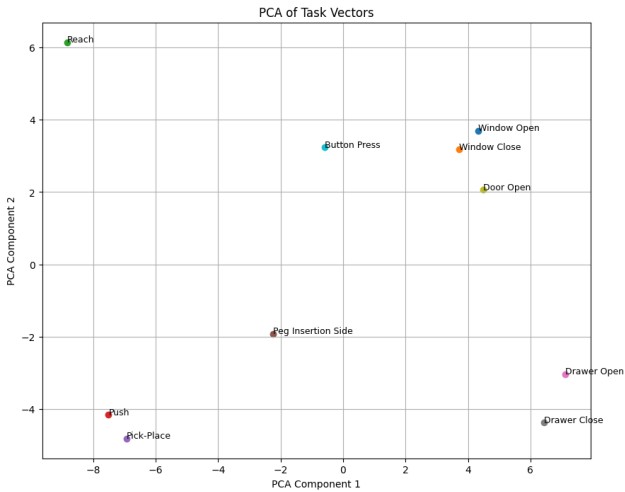

*Figure 7.* PCA for our parametrization described in 3.4.

When $K = 3$, the three Meta-World clusters (for one of the random subsamples of 30 tasks) are provided below.

On the high level, the clustering can be described as corresponding to three categories of manipulation tasks. The first cluster involves pulling and sliding manipulations (e.g., pull a puck, move end-effector, slide a plate to and from the cabinet). The second cluster includes manipulations that involve pressing and pushing (e.g., rotate the faucet, press a handle down, insert a peg into a hole, kick a soccer ball, sweep a puck). The third cluster consists of tasks involving indirect or constrained manipulation and which thus require better precision and control (e.g., bypass a wall, grasp a stick and pull or push a box with it).

| Cluster 1 | Cluster 2 | Cluster 3 |
|---|---|---|
| handle-press-side-v1 | faucet-open-v1 | hand-insert-v1 |
| peg-unplug-side-v1 | hammer-v1 | handle-pull-v1 |
| pick-out-of-hole-v1 | handle-press-side-v1 | lever-pull-v1 |
| pick-place-v1 | handle-press-v1 | peg-insert-side-v1 |
| plate-slide-back-v1 | handle-pull-side-v1 | push-wall-v1 |
| plate-slide-side-v1 | peg-insert-side-v1 | reach-wall-v1 |
| plate-slide-v1 | peg-unplug-side-v1 | shelf-place-v1 |
| push-back-v1 | pick-out-of-hole-v1 | stick-pull-v1 |
| push-v1 | pick-place-v1 | stick-push-v1 |
| reach-v1 | pick-place-wall-v1 | sweep-into-v1 |
| stick-push-v1 | plate-slide-back-side-v1 | window-close-v1 |
| | plate-slide-side-v1 | window-open-v1 |
| | plate-slide-v1 | |
| | soccer-v1 | |
| | sweep-v1 | |

*Table 9.* Meta-World Task Clusters

