# OpenReview forum: "Learning Policy Committees for Effective Personalization in MDPs with Diverse Tasks"
_ICML.cc/2025/Conference — ICML 2025 poster_

### Official Review · Reviewer_avxU · 2025-03-12

**Overall Recommendation:** 2

**Summary:**

The paper proposes PACMAN, a novel method that learns a policy committee, ensuring that at least one near-optimal policy is included for each task with high probability. The proposed approach is evaluated on Half-Cheetah Velocity, Humanoid, and MetaWorld tasks, demonstrating superior performance compared to multiple baselines.

**Claims And Evidence:**

The claims are primarily supported by empirical experiments.

**Essential References Not Discussed:**

N/A

**Experimental Designs Or Analyses:**

For comparison with baselines:
 - How is the number of parameters for PACKMAN compared with other baselines?
 - What is the optimal performance for each set of experiments? Providing this information would help in better understanding the reported performance metrics.

For experiments in metaworld,
 - Is the policy state-based or image-based?
 - What are the 30 and 20 tasks mentioned in the paper? How are they constructed? Are they randomly sampled?
 - Is the return equivalent to the success rate? Clarifying this would improve result interpretation.

**Methods And Evaluation Criteria:**

The method, especially the algorithm is not clear to me.

Regarding the clustering mechanism:
 - How many clusters are there in the committees for both experiments? Is the number of clusters predefined or learned automatically?
 - Can you provide an ablation study to analyze the impact of the number of clusters on performance?
 - Additionally, can you conduct an ablation study on the selection of key hyperparameters for the clustering mechanism?
 - Can you provide detailed analysis on what tasks are clustered together to verify the clustering mechanism works?

Regarding few-shot generalization:
 - Will new committee member be added when learning new tasks? Or will it reuse existing member?

**Other Comments Or Suggestions:**

There are some references missing in line 335 and line 346.

**Other Strengths And Weaknesses:**

N/A

**Questions For Authors:**

N/A

**Relation To Broader Scientific Literature:**

The problem is highly relevant to the field. It is a new method for multi-task learning

**Theoretical Claims:**

No I am not falimiar with the theoretical analysis.

---

> ### Author Rebuttal · Authors · 2025-03-28
>
> We thank the reviewer for the thoughtful comments and clarification questions!
>
> >**Comment:** The method, especially the algorithm is not clear to me.
>
> **Response**: The details of the algorithm are provided in Section 3.1. In summary, we cluster tasks into groups, where each group contains similar tasks (in a way that yields provable in terms of performance and generalization) and then train a policy for each group. We call the resulting set of policies a *policy committee*. This committee can then be used for both in-sample and out-of-sample multi-task RL settings, as well as for few-shot meta-RL problems.
>
> >**Comment:** Can you provide detailed analysis on what tasks are clustered together to verify the clustering mechanism works?
>
> **Response**: Certainly. At the high level, the three Meta-World clusters (for one of the random subsamples of 30 tasks) can be described as corresponding to three categories of manipulation tasks.  The first cluster involves pulling and sliding manipulations (e.g., pull a puck, move end-effector, slide a plate to and from the cabinet). The second cluster includes manipulations that involve pressing and pushing (e.g., rotate the faucet, press a handle down, insert a peg into a hole, kick a soccer ball, sweep a puck). The third cluster consists of tasks involving indirect or constrained manipulation and which thus require better precision and control (e.g., bypass a wall, grasp a stick and pull or push a box with it). We will add full details of task-to-cluster mappings in the revision.
>
> >**Comment:** How many clusters are there in the committees for both experiments?
>
> **Response**: For the main experiments, $K=3$ in all domains, but we also consider ablations with $K \in \{1,2,4\}$ (see the Appendix for details).
>
> >**Comment:** Is the number of clusters predefined or learned automatically?
>
> **Response**: The number of clusters is predefined in our experiments, but can also be obtained automatically if we have a coverage target (e.g., if we wish to cover 100% of training tasks).
>
> >**Comment:**  Will new committee member be added when learning new tasks? Or will it reuse existing member?
>
> **Response**: We reuse existing members for new tasks; in zero-shot settings, we apply these directly on out-of-sample tasks, while in few-shot settings we fine-tune policies in the committee.
>
> >**Comment:**  How is the number of parameters for PACKMAN compared with other baselines?
>
> **Response**: If an architecture used for a policy has $p$ parameters, the committee of $K$ policies will train $Kp$ parameters. Different baseline approaches vary significantly in the number of parameters $p$, as this is not typically a primary consideration in prior literature, particularly in the context of the challenging Meta-World benchmark.
>
> >**Comment:** What is the optimal performance for each set of experiments?
>
> **Response**: For meta-learning mujoco experiments, the theoretical upper bound on performance is 0. For Meta-World, the theoretical upper bound is the success rate of 1. However, we are unaware of any SOTA method that comes close to achieving either of these, and it is unknown what is a feasible optimal performance on these benchmarks with respect to out-of-sample tasks.
>
>
> >**Comment:** Is the policy state-based or image-based?
>
> **Response**: The policy is state-based.
>
> >**Comment:** What are the 30 and 20 tasks mentioned in the paper? How are they constructed? Are they randomly sampled?
>
> **Response**: We uniformly randomly partition the 50 total tasks in Meta-World into 30 trainig and 20 test tasks.
>
> >**Comment:** Is the return equivalent to the success rate? Clarifying this would improve result interpretation.
>
> **Response**:  While high returns generally correlate with a high success rate, return and success rate are not the same. Return refers to the total reward over a trajectory, which is important for the theoretical analysis as well as how RL is trained in practice (since it yields a dense reward). Success rate is the standard evaluation metric in Meta-World and is used in our experiments. We will clarify this distinction in the revised version.

---

### Official Review · Reviewer_sYFo · 2025-03-13

**Overall Recommendation:** 3

**Summary:**

The paper introduces PACMAN, a novel framework for learning policy committees in multi-task Markov decision processes (MDPs) with diverse tasks. The approach includes clustering-based approach to create a representative cover over tasks and a gradient-based alternative to group tasks based on parametric representations and then trains a separate policy per cluster. The authors support their method with theoretical guarantees and demonstrate significant empirical improvements on benchmarks such as MuJoCo and Meta-World.

**Update after rebuttal**--I appreciate the authors’ thorough revision and the additional experiments addressing my concerns. I hope those will be implemented in the revised draft. I have decided to keep my original score.

**Claims And Evidence:**

The paper makes several key claims:
- It offers provable guarantees on generalization and few-shot adaptation.
- It achieves improved performance over state-of-the-art baselines in diverse task settings.
The theoretical results (e.g., Theorems 3.7 and 3.8) are well-supported by proofs in the appendices, and the experimental results consistently back up the claims.
- However, some assumptions (e.g., tasks being parametric, Lipschitz continuity of rewards) may not hold in real-world settings.
- The proposed approaches are validated with multi-task benchmarks.

**Essential References Not Discussed:**

While the paper cites many relevant works, it might benefit from a discussion of more recent approaches to policy ensembles and adaptive task embeddings in non-parametric settings. This could help in understanding how PACMAN compares to or complements emerging methodologies. A more detailed discussion of contextual MDPs from related areas might further enrich the context. Specifically, the notion of “policy committee” feels similar to “multi-policy training” proposed in “Model-Based Transfer Learning for Contextual Reinforcement Learning (NeurIPS 2024).”

**Experimental Designs Or Analyses:**

Experiments are extensive and well-structured:

- The use of both zero-shot and few-shot evaluations provides a comprehensive picture of performance.
- Comparisons with multiple baselines help in understanding the relative strengths of PACMAN.
- The scalability of the clustering method in higher-dimensional settings and its sensitivity to the number of tasks are areas that may benefit from further clarification.
- In addition, there are many other things remained questionable in experiment settings. For instance, I’m also curious how the choice of 30 training tasks and 20 test tasks in MT-50 (Meta-World) affects the other baseline’s performance. The effect of task description and language embedding can affect to some extent.

**Methods And Evaluation Criteria:**

The methodology is thoughtfully designed:

- The clustering approach effectively handles task diversity by creating a representative cover over task parameters.
- The integration of gradient-based optimization to refine cluster representatives is innovative.
The evaluation on standard benchmarks is thorough, with comparisons to 11 state-of-the-art baselines.
- Evaluation criteria (average return of one episode or success ratio) from benchmarks seems to be reasonable.

**Other Comments Or Suggestions:**

- Section 4 → citation error in line 349
- Section 4.1 → citation error in line 335 and (EM) in line 342
- Curious how this approach relates to the special case of MoE with hard gate.
- Font size in figures are sometimes too small.
- The readability of the figures can be improved. It’s not as friendly for now.

**Other Strengths And Weaknesses:**

Strengths:
- Strong theoretical foundation with rigorous proofs.
- Extensive and convincing empirical validation across challenging benchmarks.
- Innovative combination of clustering with gradient-based refinement.

Weaknesses:
- Dependence on the assumption of parametric task structure may limit generalizability.
- Sensitivity to the hyperparameters and the challenge of tuning it in different domains.
- Scalability concerns for very high-dimensional tasks or a massive number of tasks.

**Questions For Authors:**

- How can I relate PACMAN results in Table 1 and Table 3?
- Table 4 shows that larger K could decrease the performance. Then how should we choose the optimal K in practice? any recommendations or thoughts? Similarly, I’m also curious about the tradeoff for higher epsilon mentioned in line 958.

**Relation To Broader Scientific Literature:**

The paper is well-positioned within the broader literature on multi-task and meta-reinforcement learning. It clearly distinguishes itself from prior works such as MAML, PEARL, MOORE, and task clustering approaches by addressing the challenges of task diversity and negative transfer.

**Theoretical Claims:**

The paper presents several rigorous theoretical contributions including the derivation of sample complexity bounds and few-shot guarantees. While the proofs appear sound, some assumptions (such as the parametric form of tasks and the specific Lipschitz conditions) might limit applicability in settings where such conditions are not met.

---

> ### Author Rebuttal · Authors · 2025-03-28
>
> We thank the reviewer for the insightful comments and suggestions!
>
> >**Comment:** The assumption of parametric task structure may limit generalizability.
>
> Response: We agree that assuming that tasks are parametric is a limitation. However, we show that we can often use LLM embedding of natural language task descriptions to enable parametric representations of non-parametric tasks. Our results on the Meta-World benchmark (in which tasks are non-parametric) demonstrate that this is quite effective, and suggests that our approach is nevertheless very broadly applicable. Indeed, as capabilities of LLMs continue improving, we expect the scope of applicability to continue to expand.
>
> >**Comment:**  Sensitivity to the hyperparameters and the challenge of tuning it in different domains.
>
> **Response**: Our method adds only a single hyperparameter, $\epsilon$, which is easy to tune without running RL. Our ablation studies in Appendix H suggest that choosing a small $\epsilon$ that ensures full coverage for the largest $K$ a practitioner can afford to train works well in practice. As RL algorithms (particularly in multi-task settings) come with a large number of hyperparameters that are difficult to tune in practice, we view the addition of the $\epsilon$ hyperparameter to be well worth the performance improvement and theoretical guarantees.
>
> >**Comment:**  Scalability concerns for very high-dimensional tasks or a massive number of tasks.
>
> **Response**: While theoretically our framework is hard in high-dimensional settings, in practice the computational cost is dominated by RL (seconds or minutes for clustering compared to hours for RL). Moreover, we show that the proposed gradient-based approach works well in higher dimensions. Moreover, both our greedy and gradient-based approaches run in polynomial time in the number of tasks. Nevertheless, consideration of extremely high dimensions and massive numbers of tasks remain important research challenges in multi-task and meta-RL.
>
> >**Comment:**  ...A more detailed discussion of contextual MDPs might further enrich the context... The notion of “policy committee” feels similar to “multi-policy training” proposed in “Model-Based Transfer Learning for Contextual Reinforcement Learning (NeurIPS 2024).”
>
> **Response**: Thank you for pointing this out! We will add a discussion of contextual MDPs in the revision. While our notion of a “policy committee” is similar to “multi-policy training”, the focus of Cho et al. is on a sequential task selection problem, with a task-specific policy trained on the selected task. Our setting, in contrast, begins with a given set of tasks (we cannot choose tasks for training) and entails clustering, with the policy committee associated with task clusters. We will add this discussion in the revision.
>
> >**Comment:** How can I relate PACMAN results in Table 1 and Table 3?
>
> **Response**: As Table 1 is for few-shot and Table 3 for zero-shot generalization, they are not directly comparable. The former shows that PACMAN outperforms SOTA in few-shot learning, while the latter shows that the novel clustering approach we propose for multi-task learning is better than using conventional clustering methods such as Kmeans++ for task clustering in this setting.
>
> >**Comment** Table 4 shows that larger K could decrease the performance. Then how should we choose the optimal K in practice? any recommendations or thoughts? Similarly, I’m also curious about the tradeoff for higher epsilon mentioned in line 958
>
>
> **Response**: We discuss this in detail in section H.2.2 (the paragraph under the table in question). In practice, our guideline is to use $K$ no larger than what is sufficient to provide a $(\epsilon,1)$ cover (i.e., cover all training tasks); moreover, $K=3$ has worked well across distinct domains and in many practical settings may thus be a good starting point. As for $\epsilon$, we provide some guidance on its selection in our response to Reviewers fviP and zPNZ given a fixed $K$.
>
>
> >**Comment:** Curious how this approach relates to the special case of MoE with hard gate.
>
>
> **Response**: In MoE, the gating mechanism typically needs to be trained as part of the network, and with hard gating, selecting the right expert is a challenging task. In contrast, our method alleviates this computational burden by directly producing a set of policies, and the best can be quickly identified through policy evaluation for any task.
>
> >**Comment:** Citation error and figure readability.
>
> **Response**: Thank you for pointing this out! We will rectify the citation errors and improve the readability of the figures in the revised version.

---

### Official Review · Reviewer_zPNZ · 2025-03-18

**Overall Recommendation:** 2

**Summary:**

The paper introduces a new learning paradigm, called policy committee learning effectively for solving multi-task RL. More specifically, a policy committee targets learning a set of policies maximizing the best-performing policy's discounted return for any given task sampled from a predefined task distribution. The paper assumes that the tasks differ by their transition dynamics and reward functions, but not state and action spaces. Also, the paper assumes access to a parameterization of these tasks.

For this new learning paradigm, the paper introduces three algorithms, GEA, GIA, and, Gradient-based Coverage. These methods centers around finding a clustering that achieves $(\epsilon, 1 − \delta)$-parameter-cover following Definition 3.1. Further theoretical analysis indicates that learning individual policy for each cluster leads to a solution to policy committee learning (Definition 2.1) with bounded optimality gap.

The paper conducts the experiments in MetaWorld, Halfcheetah, and Humanoid domains, and demonstrating improved asymptotic performance and few-shot learning performance compared to recent meta-learning and multi-task RL approaches.

**Claims And Evidence:**

The paper claims a theoretically-grounded framework for learning policy committees, however the theory is rather limited in that it assumes all the tasks share the same transition dynamics. Also, I am not sure whether the theoretical bound is tight enough can explain the performance gain in the experiments.

**Essential References Not Discussed:**

The idea is pretty similar to mixture of expert for RL:
Celik, Onur, Aleksandar Taranovic, and Gerhard Neumann. "Acquiring diverse skills using curriculum reinforcement learning with mixture of experts." arXiv preprint arXiv:2403.06966 (2024).

In Onur et. al., the clustering (partition of the task space) is guided by the experts performance, e.g., if the expert shows promises in solving the task. The paper instead considers the $(\eplison, 1 − \delta)$-parameter-cover. I think the author should consider differentiating with Onur et. al.

**Experimental Designs Or Analyses:**

1. MetaWorld domain breaks the assumption made by the theoretical analysis that all the tasks share the same transition dynamics, as different tasks involve interacting with different objects, and therefore uses different transition dynamics.

2. It is also a bit suspicious that although the language embedding provide a continuous parameterization of the tasks, the real task space is discrete and only have 50 tasks in total. In that, it is hard to believe simpler clustering strategies, like KMean++, do not give the same clustering result. I saw that comparison with other clustering algorithms is only conducted in HalfCheetahVel. Could author also conduct comparison in MetaWorld domain?

3. It would be interesting to visualize the clustering results in HalfCheetahVel and compare it with KMean++. HalfCheetahVel uses a simple one-dimensional task parameter, I wonder in what degree the clustering result of PACMAN differs from conventional goal of minimizing sum of shortest distances.

**Methods And Evaluation Criteria:**

Yes, learning a policy committee makes sense for solving multi-task RL problems. Learning different policies may mitigate the gradient conflict and also by limiting the total number of policies, the learning may become more efficient benefit from the share similarity between tasks.

**Other Comments Or Suggestions:**

1. There are a couple of missing citations: (1) line 336 SAC; (2) line 349 MetaWorld.
2. In Theorem 3.4, 3.7, $1/e$ is not defined. In Theorem 3.8, $H$ is not defined. The theoretical analysis could be presented for clearly.

**Other Strengths And Weaknesses:**

N/A

**Questions For Authors:**

1. In theorem 3.7, could the author give an rough estimation of the optimality bound for HalfCheetch domain? Specifically, what are the magnitudes for
$2L\frac{1−\gamma^{h+1}}{1−\gamma} \epsilon + \eta$ and $(1 − \frac{1}{e}) )(1 − \delta^∗(K, \epsilon) − K\beta))$, using $K=3$ and $\epsilon = 0.6$. How does the bound compared to $K=1$ using one single policy to learn all the tasks?
2. It is also a bit suspicious that although the language embedding provide a continuous parameterization of the tasks, the real task space is discrete and only have 50 tasks in total. In that, it is hard to believe simpler clustering strategies, like KMean++, do not give the same clustering result. I saw that comparison with other clustering algorithms is only conducted in HalfCheetahVel. Could author also conduct comparison in MetaWorld domain?
3. Could the author also provide the actual assignments of the 50 tasks to the three policies in the policy committee?
4.  Could the authors also provide the visualization of the clustering results in HalfCheetahVel and compare it with KMean++. HalfCheetahVel uses a simple one-dimensional task parameter, I wonder in what degree the clustering result of PACMAN differs from conventional goal of minimizing sum of shortest distances.

**Relation To Broader Scientific Literature:**

The paper introduces a new learning paradigm called policy committee learning. I think the idea is close to mixture of experts.

**Theoretical Claims:**

I did not check the proofs.

---

> ### Author Rebuttal · Authors · 2025-03-28
>
> We thank the reviewer for the thoughtful comments and suggestions!
>
> >**Comment:** Meta-World domain breaks the assumption made by the theoretical analysis that all the tasks share the same transition dynamics, as different tasks involve interacting with different objects, and therefore uses different transition dynamics.
>
>
> **Response**: This was by design. Our key finding is that while our analysis (like most worst-case analysis) makes restrictive assumptions, its value is exhibited by the Meta-World experiments that demonstrate robustness of the proposed approach to the assumptions. This is common in theoretical analysis, which of necessity must make assumptions that are inevitably violated in practice; what ultimately demonstrates value is empirical performance. The theoretical analysis, in turn, serves both to explain why we should expect strong performance in practice, and to ensure that the approach is principled (and, therefore, less likely to be fragile as we change domains).
>
> Thus, we view the combination of (a) superior empirical performance on the mainstream benchmarks and (b) provable worst-case guarantees in an important and broad array of settings (diverse rewards) makes our approach a particularly significant advance over prior art.
>
>
> >**Comment:** It is hard to believe simpler clustering strategies, like KMeans++, do not give the same clustering result... Could author also conduct comparison in Meta-World domain?
>
> **Response**: We indeed obtained different clusters using PACMAN than using Kmeans++ and, as a result, much better performance for Meta-World, as shown in the table below:
>
> |       | Kmeans++    |  PACMAN (Ours)     |
> |-------|--------|--------|
> |125K Steps| 0.22 $\pm$ 0.06 | 0.36 $\pm$ 0.07|
> |250K Steps| 0.30 $\pm$ 0.07 | 0.48 $\pm$ 0.09|
>
> Notably, PACMAN exhibits $\sim 60$\% performance impovement over Kmeans++.
>
> >**Comment**: It would be interesting to visualize the clustering results in HalfCheetahVel and compare it with KMean++.
>
> **Response**: In Appendix H.2.1., we provided a histogram of the final clustering results in terms of individual task returns. We will also include a visualization of our and KMeans++ clusters in the revision.
>
> >**Comment**: Differentiating the work from *Onur et. al., ICML 2024*
>
> **Response:** While our approach shares the key insight of addressing task diversity with MoE methods like the one proposed by Onur et al., we take it a step further, generating a small number of clusters upfront and using RL for each cluster independently thereafter. This facilitates more efficient and effective clustering (since no RL is involved for that) with provable guarantees, and is particularly valuable for few-shot adaptation (also with theoretical guarantees). Notably, our baselines include MoE-based methods (e.g., MOORE from ICLR 2024).
>
> >**Comment:**  Could the author give an rough estimation of the optimality bound for HalfCheetch domain?
>
> **Response:** $2L\frac{1-\gamma^{h+1}}{1-\gamma} \epsilon \approx 100$, which is quite tight in this setting. When $K=3$, the committee generated by GIA yields near-perfect coverage $1-\delta^* \approx 1$, whereas for $K=1$, $1-\delta^* \approx 0.4$. The second bound is thus $\sim 0.63$ for the former and $\sim 0.25$ for the latter, so that the advantage of $K=3$ is a factor of 2.5. In practice, we find that these **worst-case** bounds are quite conservative, and empirical performance is far better.
>
> >**Comment:** I checked Appendix J, but I would be interested to see the actual assignment of the 50 tasks to the three policies in the policy committee. Could the author also provide the actual assignments of the 50 tasks to the three policies in the policy committee?
>
> **Response:** We will add full details of task-to-cluster mappings in the revision. At the high level, the three Meta-World clusters (for one of the random subsamples of 30 tasks) can be described as corresponding to three categories of manipulation tasks.  The first cluster involves pulling and sliding manipulations (e.g., pull a puck, move end-effector, slide a plate to and from the cabinet). The second cluster includes manipulations that involve pressing and pushing (e.g., rotate the faucet, press a handle down, insert a peg into a hole, kick a soccer ball, sweep a puck). The third cluster consists of tasks involving indirect or constrained manipulation and which thus require better precision and control (e.g., bypass a wall, grasp a stick and pull or push a box with it).
>
> >**Comment:** There are a couple of missing citations: (1) line 336 SAC; (2) line 349 MetaWorld. In Theorem 3.4, 3.7 missing defintions
>
> **Response:** Thank you for pointing this out. We will rectify this in the revision.

---

### Official Review · Reviewer_fviP · 2025-03-20

**Overall Recommendation:** 3

**Summary:**

This paper introduces PACMAN, a novel framework and algorithmic approach for learning policy committees in multi-task reinforcement learning (MTRL) and meta-reinforcement learning (meta-RL) settings with diverse tasks. The key challenge addressed is the difficulty of generalizing effectively across diverse tasks, where traditional MTRL and meta-RL approaches often struggle due to negative transfer or computational expense.

PACMAN introduces a theoretically grounded approach for learning policy committees. It frames the problem as finding an (ε, 1-δ)-cover of the task distribution, ensuring that with high probability, there's at least one near-optimal policy in the committee for any encountered task.

The paper presents two practical algorithms: Greedy Intersection Algorithm (GIA), which provides provable approximation and task sample complexity guarantees when task dimensions are low and Gradient-Based Coverage, which is a general, practical, gradient-based approach suitable for higher-dimensional tasks.

The paper provides a provable sample complexity bound for few-shot learning that depends only on the number of clusters (committee size) and not on the state or action space size.

The authors performed extensive experiments on MuJoCo and Meta-World benchmarks demonstrate that PACMAN outperforms state-of-the-art MTRL, meta-RL, and task clustering baselines in training, generalization, and few-shot learning, often by a large margin.

**Claims And Evidence:**

The claims are generally well-supported. The theoretical results (Theorems 3.3, 3.4, 3.5, 3.7, 3.8, Lemma 3.6) have provided proofs (although some are deferred to the appendix). The empirical results are comprehensive, covering multiple environments, baselines, and evaluation settings. The ablation studies (varying K and ε) provide further evidence for the effectiveness of the proposed approach.

**Essential References Not Discussed:**

No

**Experimental Designs Or Analyses:**

The experimental design is generally sound. The choice of environments, baselines, and evaluation metrics is appropriate. The ablation studies help to understand the impact of key hyperparameters. The use of multiple random seeds and reporting of standard deviations provides statistical confidence in the results.

**Methods And Evaluation Criteria:**

The methods and evaluation criteria are appropriate. The use of MuJoCo and Meta-World benchmarks is standard in the field. The evaluation metrics (training performance, zero-shot generalization, few-shot adaptation) are relevant to the problem. The comparison to a wide range of baselines strengthens the evaluation.

**Other Comments Or Suggestions:**

None

**Other Strengths And Weaknesses:**

Strengths:
1. Strong theoretical grounding with provable guarantees.

2. Novel algorithmic approach combining clustering and policy committee learning.

3. Empirical results demonstrating significant improvements over strong baselines.

4. Applicability to both parametric and non-parametric task settings.

5. Clear and well-written presentation.

Weaknesses:

1. The reliance on a parametric task representation (or the ability to obtain one using LLMs) is a limitation, although a common one.

2. The need to tune the hyperparameter ε is a practical consideration.

3. The computational cost of training multiple policies could be a concern in some settings, although the paper argues that the RL step dominates.

4. The GIA algorithm's exponential time complexity in d makes it only efficient for constant d.

**Questions For Authors:**

1. The paper mentions that the RL step will typically dominate computational complexity. Could you provide some empirical evidence or a more detailed analysis comparing the time spent on clustering versus RL training, particularly for the Meta-World experiments? This would affect evaluation by giving insight into computational cost.

2. In the Meta-World experiments, you use MOORE for within-cluster training. Have you experimented with other MTRL or meta-RL algorithms for this step, and if so, how did they compare? This would clarify the extent to which the gains are attributable to the overall framework versus the specific choice of within-cluster RL method.

3. The (ε, 1-δ)-cover concept is central to the theoretical analysis. Could you elaborate on the practical implications of choosing different values of ε? How does one typically set this hyperparameter in a new environment? Clarifying how users could tune this parameter.

**Relation To Broader Scientific Literature:**

The key contributions build on MTRL and meta-RL literature, improving over baselines like RL2 and VariBAD. The policy committee idea relates to ensemble methods or mixture of experts, and clustering is common in ML, but applying it to RL policy learning is novel. Theoretical bounds are crucial for RL efficiency.

**Theoretical Claims:**

I have roughly checked the proofs provided in the main text and skimmed through proofs in the appendix, and they appear to be logically sound, given the assumptions made.

---

> ### Author Rebuttal · Authors · 2025-03-28
>
> We thank the reviewer for the thoughtful comments!
>
>
> >**Comment:** The reliance on a parametric task representation (or the ability to obtain one using LLMs) is a limitation, although a common one.
>
> **Response**: Indeed, our method does require access to a parameterization for the tasks. We note, however, that our results with embeddings based on language descriptions suggest that our approach can nevertheless be very effectively applied to a very broad array of problems. Indeed, the continuing improvement of both the quality and scope of LLMs suggests that the space of problems to which our approach can be effectively applied will likely continue to expand.
>
> >**Comment:** The need to tune the hyperparameter ε is a practical consideration.
>
> **Response**: Our method introduces only one hyperparameter, $\epsilon$, which is easy to tune, particularly in comparison to myriad of hyperparameters that typical deep RL methods introduce. For example, our ablation studies in Appendix H suggest that choosing the smallest $\epsilon$ that ensures full coverage for the largest $K$ a practitioner can afford to train works well in practice.
>
> >**Comment:** The GIA algorithm's exponential time complexity in d makes it only efficient for constant d.
>
> **Response**: This is true; indeed, as we show in Theorem 3.3, the problem is inapproximable in the worst case.  We address this issue by presenting a practical gradient-based coverage algorithm, which is the main workhorse of the Meta-World experiments, and which outperforms SOTA baselines.
>
> >**Comment:** The paper mentions that the RL step will typically dominate computational complexity. Could you provide some empirical evidence or a more detailed analysis comparing the time spent on clustering versus RL training, particularly for the Meta-World experiments? This would affect evaluation by giving insight into computational cost.
>
>
> **Response**: For the Meta-World experiments, training a single policy requires approximately 40 hours using an A40 GPU for 1M steps. In comparison, clustering takes about 1 second in a Google Colab notebook, and obtaining task embeddings takes around 2 minutes on an A40 GPU. We will include these details along with the corresponding timings in the revised version of the paper.
>
>
> >**Comment:** In the Meta-World experiments, you use MOORE for within-cluster training. Have you experimented with other MTRL or meta-RL algorithms for this step, and if so, how did they compare? This would clarify the extent to which the gains are attributable to the overall framework versus the specific choice of within-cluster RL method.
>
> **Response:** While we have not extensively evaluated this issue, we did run experiments with alternative within-cluster training to MOORE. We have found that PACMAN typically outperforms the approach it wraps by a considerable margin. For example, PACMAN trained using CMTA produced significant improvement as well:
>
> |       |   CMTA   | PACMAN (CMTA)  |
> |-------|--------|--------|
> |125K Steps| .20 $\pm$ .08 | .28 $\pm$ .08|
> |250K Steps| .25 $\pm$ .09 | .41 $\pm$ .10|
>
>
> >**Comment:** The (ε, 1-δ)-cover concept is central to the theoretical analysis. Could you elaborate on the practical implications of choosing different values of ε? How does one typically set this hyperparameter in a new environment? Clarifying how users could tune this parameter.
>
> **Response**:  In Appendix H.2.2, Table 5, we show that when the coverage is not complete due to a small ε, performance significantly degrades.  Practically, the choice of ε depends on the cost of training a policy. A user should aim to select the largest committee size they can afford and then choose an ε that ensures maximum coverage. One approach is to first compute the distances between all tasks' embeddings. This could provide an initial estimate for setting ε. From there, we can fine-tune ε to ensure it is small while still guaranteeing adequate coverage. From what we have observed, ε is far less sensitive than typical RL hyperparameters.

---

### Decision · Program_Chairs · 2025-05-01

**Decision:**

Accept (poster)

**Comment:**

The reviewers generally agreed that the paper provided strong theoretical contributions that support solid empirical results on two benchmarks (MuJoCo and Meta-World). Overall, the paper is clear, well-motivated, and offers impressive improvements over existing methods. There were some concerns regarding the method relying on parametric task representations (a common issue) and sensitivity to hyperparameters. After reading the paper myself, I also agree with one reviewer’s comment that the paper would benefit from a more explicit algorithm summarizing the entire approach even in the appendix — that would clarify exactly how all of the parts developed over the paper fit together into the final approach.